# The king's spice cabinet–Plant remains from *Gribshunden,* a 15<sup>th</sup> century royal shipwreck in the Baltic Sea

**Mikael Larsson**[1]*, **Brendan Foley**[1,2]

**1** Department of Archaeology and Ancient History, Lund University, Lund, Sweden, **2** Blekinge County Museum, Karlskrona, Sweden

* mikael.larsson@ark.lu.se

## Abstract

Maritime archaeological investigations of the wreck of the medieval warship *Gribshunden* (1495), flagship of King Hans of Denmark and Norway, have revealed diverse artifacts including exotic spices imported from far distant origins: saffron, ginger, clove, peppercorns, and almond. The special circumstances of the vessel's last voyage add unique context to the assemblage. *Gribshunden* and an accompanying squadron conveyed the king, courtiers, noblemen, and soldiers from Copenhagen to a political summit in Kalmar, Sweden. At that conference, Hans expected the Swedish Council to elect him king of Sweden, and thereby fulfill his ambition to reunify the Nordic region under a single crown. To achieve this, Hans assembled in his fleet and particularly aboard his flagship the people and elite cultural signifiers that would convince the Swedish delegation to accept his rule. Along the way, the ships anchored near Ronneby, Blekinge. Written sources record that an explosion and fire caused *Gribshunden* to sink off Stora Ekön (Great Oak Island). Exotic spices were status markers among the aristocracy in Scandinavia and around the Baltic Sea during the Middle Ages (1050–1550 CE). Until the *Gribshunden* finds, these extravagances have rarely or never been represented archaeologically. Evidence of their use and consumption in medieval Scandinavia has been limited to sparse written references. We present here the botanical remains from the *Gribshunden* shipwreck and compare them to previous archaeobotanical finds from the medieval Baltic region. These opulent status symbols traveled with a medieval king en route to a major historical event. The combination of textual and archaeological evidence allows a novel analytical view of the social environment in which these luxurious foods were consumed.

## Introduction

On the southern coast of Sweden, among the islands of the Blekinge archipelago, sits the wreck of a late medieval royal warship variously known as *Gribshunden* or *Griffen* (Fig 1). Built in 1485, probably in the Low Countries [1], the vessel was a "floating castle" that served as the flagship and mobile seat of government for King Hans of Denmark and Norway. Just before

**Data Availability Statement:** All relevant data are within the paper and its Supporting Information files.

**Funding:** This research was supported by funds awarded to Brendan Foley from Crafoord

Foundation, Sweden [grant numbers 20190008 and 20200003], the Swordspoint Foundation, USA [no grant number], and the Huckleberry Foundation, USA [no grant number], and by funds awarded to Mikael Larsson from the Swedish Research Council [grant number 2019-02547], supplemented by internal funding from Blekinge Museum and the Department of Archaeology and Ancient History, Lund University. Brendan Foley received salary from the Swordspoint Foundation, USA [no grant number], the Huckleberry Foundation, USA [no grant number], Blekinge Museum and the Department of Archaeology and Ancient History, Lund University. Mikael Larsson received salary from the Swedish Research Council [grant number 2019-02547]. The funders had no role in study design, data collection and analysis, decision to publish, or preparation of the manuscript.

**Competing interests:** The authors have declared that no competing interests exist.

*midsommar*, at the end of June 1495, the ship anchored off Stora Ekön (Great Oak Island) [2]. Hans disembarked, presumably with a noble retinue to meet officials in the nearby town of Ronneby. While he was ashore, a fire and explosion claimed the ship. The hull settled onto a soft sea floor in about 11 meters of water. Some salvage undoubtedly occurred immediately, as the masts and rigging and perhaps even superstructure stood above the sea surface. Eventually, the upper works collapsed and fine-grained sediments infilled the hull, preserving the archaeological deposit.

The circumstances of the ship's final voyage are noteworthy. *Gribshunden* sank while Hans sailed with it and a fleet to a political summit in Kalmar, Sweden to meet with the Swedish regent and noble council. At the culmination of the five-week negotiation, Hans expected the Swedish Council to elect him as their king, thereby fulfilling his ambition to re-unite the Nordic region under a single monarch. To achieve this result, Hans would demonstrate to the Swedish delegation the authority and wealth of his crown. He carried with him all manner of power displays: his warships, shipboard artillery, a battalion of professional soldiers, and small arms including crossbows and gunpowder weapons. Buttressing these hard power elements were soft power signifiers: coinage, artwork, splendid livery, and delicacies for feasting. Archaeological investigations of the site have produced material evidence of all of these facets of Hans' strategy [3–7].

This study presents macrofossil botanical finds from the 2021 excavation campaign, which revealed a diverse assemblage of plant materials, including exotic flavorings with no Nordic-region archaeological precedent: saffron, cloves, and ginger. The same context produced peppercorns, mustard, caraway, dill, raspberry and blackberry, cucumber, grape, almonds, and hazelnuts. Preservation of these organic remains is due to the Baltic Sea's exceptional environmental conditions. The Baltic is well-known for preserving archaeological material, particularly wooden shipwrecks. At the *Gribshunden* wreck site, this is due to low salinity of about 7.7 Practical Salinity Units, combined with low temperatures averaging 9 degrees Celsius (seasonal range 2–19° C) [5,6]. These factors prevent the propagation of the wood-eating *Teredo navalis* shipworm [8,9]. The wooden s lhipwreck structure on the sea floor creates a microenvironment by capturing drifting marine algae, with seasonal deposits of algae reaching depths of 40 cm in and around the wreck. As the algae decays, localized areas of oxygen depletion occur, characterized by the presence of white mats of organic matter. These factors contribute to excellent preservation of archaeological remains, particularly plant foods carried aboard the ship: cereals, oilseeds, fruits, vegetables, spices, nuts, and berries. All have been recovered from this site and identified through archaeobotanical research.

This is the first report on archaeobotanical material from the wreck site. We present all economic plant species found on this late medieval ship and archaeobotanical references for each taxon are summarized. The geographical frame of this investigation is centered on Scandinavia and the Baltic Sea area: we refer to this as the "Baltic region". The temporal focus is the Middle Ages. In Scandinavian historical archaeology, this is typically divided into the Early Medieval period (1050–1300 CE) and the Late Medieval period (1300–1550 CE).

The plant remains from the wreck site are important for two reasons. First, the diversity of edible plants includes exotic species rarely or never found in medieval archaeological contexts, and most specimens from the wreck exhibit an excellent degree of preservation condition. The use of some of these spices in northern Europe is known only from medieval written sources. The *Gribshunden* collection of spices represents the earliest archaeological examples for several of these luxury goods in the Baltic region, and in Northern Europe [10–13]. Second, the archaeological context of the plant remains is the sterncastle of the royal flagship. Here the archaeobotanical assemblage is found in an area accessible to certain sailors, such as the helmsmen, but hypothesized to have housed only senior officers and royal/noble passengers. This

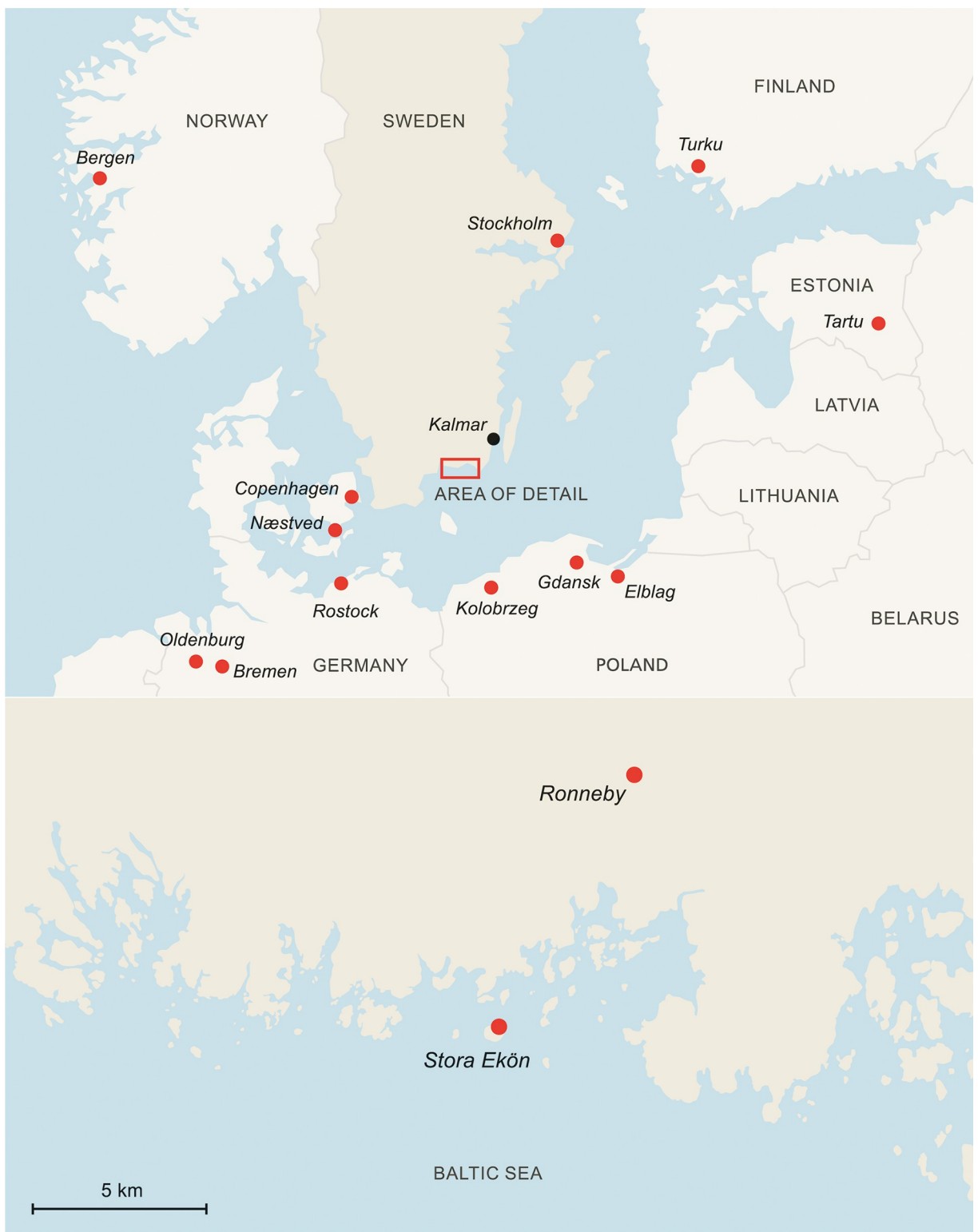

**Fig 1. Location of the *Gribshunden* shipwreck.** Map depicting the wreck site in southeastern Sweden (red quadrant), and showing the main medieval towns along the Baltic coasts with archaeobotanical data discussed in this paper. Area of detail show wreck site located north of Stora Ekön island. Republished from Media Tryck, Lund University under a CC BY license, with permission from Frida Nilsson, original copyright 2022.

presents an opportunity to learn more about the social environments in which luxury goods were consumed in late medieval northern Europe. Artifacts recovered from the loci immediately surrounding the spices represent possessions of the king and noblemen. Finds include a purse of silver coins, armor, weapons, a cask containing a butchered sturgeon, and a wooden tankard incised with a crown symbol [3,14]. The newly-discovered stores of exotic foods and spices attest to other extravagances.

The *Gribshunden* assemblage of provisions and exotic spices is the only known archaeological example of a substantially complete royal medieval pantry. The archaeobotanical data and the context of the botanical finds provide an unparalleled insight into the workings of the late medieval Nordic royal court. We glimpse the cultural and social signifiers of the elite: how the monarch and nobility behaved, what they ate, how their food was prepared. This allows discussion of medieval elite consumption and the social differentiation of foodways.

Previous marine archaeobotanical work has identified plant remains from a few shipwrecks in Northern Europe, notably the 1545 wreck of Henry the VIII's warship *Mary Rose* and the 15th century Copper Wreck near Gdansk [15–17]. These studies contribute information about trade and commodity consumption, victualling at sea, and life on board ship. The botanical finds from these wrecks are compared and discussed in relation to the new finds from *Gribshunden*. These recent botanical discoveries are of some relevance to the experience of the medieval maritime world, but more generally provide insights to the use of luxury foods in the highest echelons of medieval society in the Baltic region.

## Archaeological background

The wreck at Stora Ekön was re-discovered by sport scuba divers in the 1960-70s. Around 2000 a diver recognized the possibility that it was a medieval shipwreck, based on the distinctive oak gun carriages that once held wrought iron artillery. The Blekinge County Administrative Board then engaged a maritime archaeological reconnaissance of the wreck. This showed the main wreck site is approximately 30 m long and 10 m wide, with coherent structure underlying a mass of disarticulated and semi-articulated timbers. Between 2000 and 2012 archaeologists recovered nine oak gun carriages, collected thirteen dendrochronological samples from various structural elements, performed a limited test excavation along the centerline at the stern of the ship, and conducted an acoustic survey over the site. Concurrent with these archaeological activities, a naval historian identified the wreck as *Gribshunden* [18–21].

After a multi-year hiatus, a new research initiative was launched in 2019 and a second limited excavation trench was opened. This straddled the starboard side of the hull slightly aft of amidships, revealing hull structure and producing 60 artifacts and samples [22]. Activities in 2020 and 2021 consisted of a metal detection survey, recovery of wooden cask components for dendrochronology and analytical chemistry, and a third excavation adjacent to the 2019 trench [5,6,23].

## Materials and methods

### Maritime archaeological excavation and sampling

Archaeological investigation in and around the locus of the archaeobotanical remains discussed here took place over two excavation campaigns (Fig 2). In 2019, a trench measuring 2 x 6 x 1.5 m was excavated forward of the archaeobotanical sampling locus. No archaeobotanical investigation was conducted during that excavation. In 2021, a trench measuring 2 x 3 x 1.2 m was placed aft and inboard of the 2019 trench, separated from it by about 1.3 m. Diving archaeologists employed standard excavation tools and methods, including a venturi water dredge for removal of suspended sediments. After each excavation rotation, a mesh catchment bag

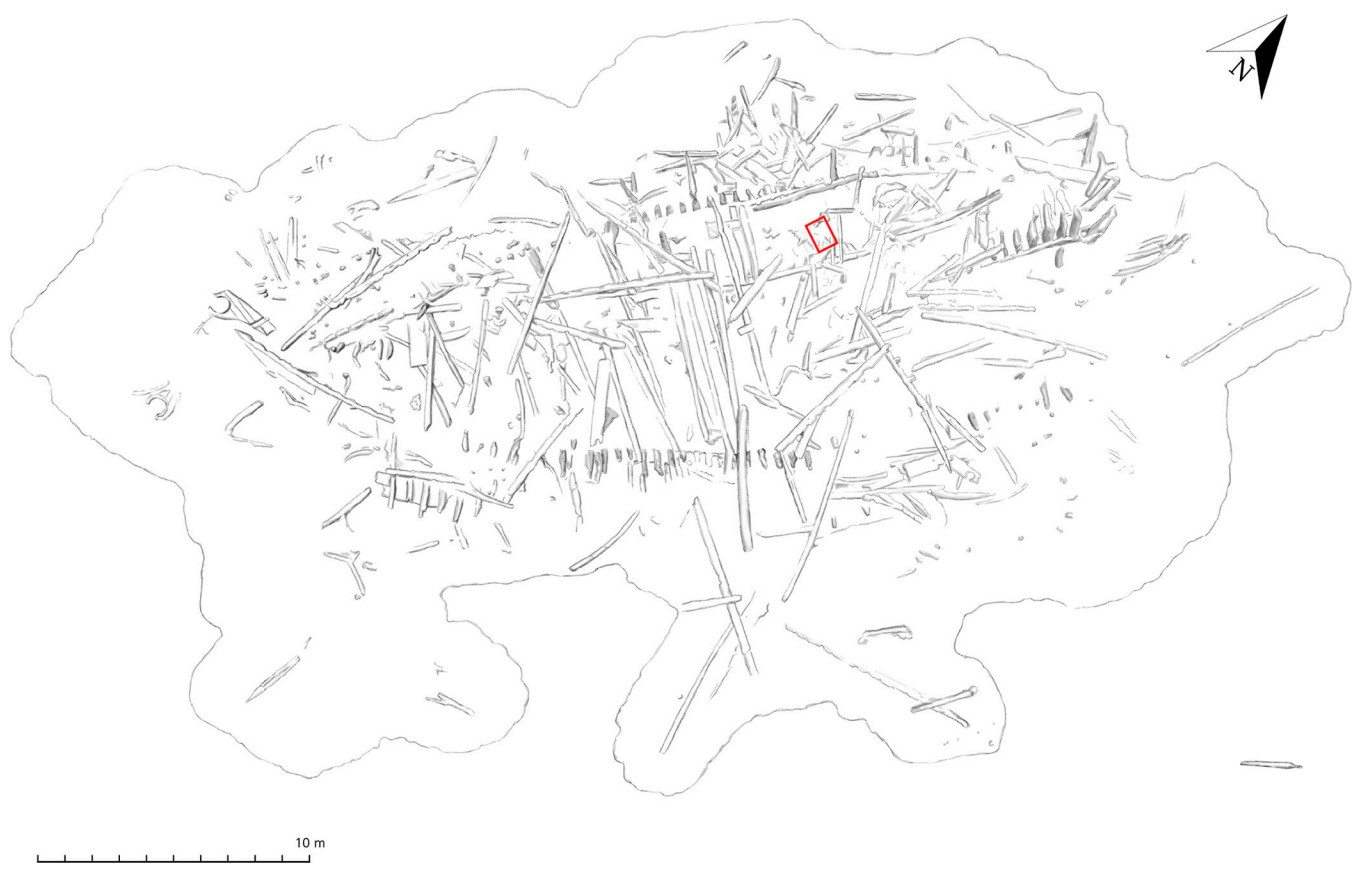

10 m

**Fig 2. Three-dimensional model of the wreck site.** The model produced during the 2019–21 field campaigns illustrates the *Gribshunden* shipwreck site. The square identifies the outline of the excavation trench from where the archaeobotanical samples were collected. Republished from Media Tryck, Lund University under a CC BY license, with permission from Frida Nilsson, original copyright 2022.

attached to the water dredge discharge was recovered and sifted on the deck of the surface support platform. The material captured in the catchment bag from a locus in the western corner of the trench included almond shell and peppercorns. Subsequent excavation in that locus delivered more almonds and peppercorns, and revealed deposits of saffron directly underneath and adjacent to a deck beam, in a stratum 5–10 cm below the sediment surface. Archaeological divers collected the saffron deposits by hand, and observed several additional discrete concentrations of archaeobotanical remains in the trench balk. They collected sediment block samples from this locus for archaeobotanical analysis, placing them in plastic containers. The saffron was recovered in a water-filled plastic sample bag. These samples were stored initially at Blekinge Museum and later shifted to the Department of Archaeology and Ancient History, Lund University. At the conclusion of the excavation campaign, the trench balk at the spice locus was reinforced with casks staves that originated from a position directly adjacent to the spices, with the intention of continuing excavation and sample recovery in that locus in future interventions [6].

All necessary permits were obtained for the described study, which complied with all relevant regulations. The 2021 maritime excavation campaign was conducted under permit 431-1299-2021 (permit holder: Blekinge Museum; project archaeological director Brendan Foley) authorized by *Länsstyrelsen i Blekinge*, Sweden (Blekinge County Administrative Board). The collected and studied botanical specimens are curated at the Department of Archaeology and

Ancient History, Lund University (address Helgonavägen 3, 223 63 Lund, Sweden, for contact: Dr. M Larsson).

## Processing and laboratory methods

The botanical remains recovered from *Gribshunden* originate from four sediment block samples of 0.5–2.0 liter each. All samples were processed by floatation with running water, and sieved over three mesh sizes: 1 mm, 0.4 mm and 0.25 mm. The saffron stigma masses were collected by hand by divers, and the deposits were placed in a 5-liter bag during the dive. This was not subject to wet-sieving; the "water-sample" (sample 5, Table 1) contained the

**Table 1. List of economic plant species in the archaeobotanical plant remains from the *Gribshunden* wreck site.**

| | | Sample no. | 1 | 2 | 3 | 4 | 5 |
|---|---|---|---|---|---|---|---|
| | | Sample vol. (liter) | 2,0 | 2,0 | 1,6 | 0,5 | water sample |
| **Cereal grain** | | Botanical remains | | | | | |
| Triticum cf. aestivum | Bread wheat | testa | · | · | 1 | · | · |
| Triticum sp. | Wheat indet. | testa | · | 2 | · | · | · |
| **Oilseeds** | | | | | | | |
| Linum usitatissimum | Flax | seed | · | · | 1 | · | · |
| **Fruits and vegetables** | | | | | | | |
| Cucumis sativus | Cucumber | seed | · | 1 | · | · | · |
| Vitis vinifera | Grape | seed | · | 1 | · | · | · |
| Rubus fruticosus | Blackberry | seed | · | 1 | 1 | · | · |
| Rubus idaeus | Raspberry | seed | · | · | 1 | · | · |
| Berry, indet. | | seed | · | · | 1 | · | · |
| **Herbs and spices (and medicinal plants)** | | | | | | | |
| Anethum graveolens | Dill | fruit | · | 1 | 2 | · | · |
| Brassica nigra | Black mustard | seed | 7 | 82 | 33 | 4 | · |
| Carum carvi | Caraway | fruit | 1 | · | 2 | · | · |
| cf. Carum carvi | Caraway | fruit, mesocarp missing | 3 | 1 | 4 | · | · |
| Crocus sativus | Saffron | stigmas | √ | √ | √ | √ | 450 ml |
| Crocus sativus | Saffron | ground stigmas (powder conglomerates) | · | · | √ | · | · |
| Humulus lupulus | Hop | fruit | 1 | · | · | · | · |
| Hyoscyamus niger | Henbane | seed | · | 1 | · | · | · |
| Piper nigrum | Black pepper | fruit (peppercorn) | 179 | 979 | 658 | 139 | · |
| Piper nigrum | Pepper | stalk segment | 1 | 9 | 6 | · | · |
| Syzygium aromaticum | Clove | whole flower bud | · | 5 | 3 | 1 | · |
| Syzygium aromaticum | Clove | hypanthium, complete/near complete | 8 | 32 | 14 | 7 | · |
| Syzygium aromaticum | Clove | hypanthium, half or less | 23 | 88 | 62 | 27 | · |
| Syzygium aromaticum | Clove | globular head | 5 | 17 | 11 | · | · |
| Syzygium aromaticum | Clove | petals | 9 | 15 | 24 | 3 | · |
| Syzygium aromaticum | Clove | ovary, complete/near complete | 18 | 42 | 43 | 11 | · |
| Syzygium aromaticum | Clove | stalk segment | 1 | 8 | 9 | 1 | · |
| Zingiber officinale | Ginger | rhizome epidermis parts | 5 | 34 | 23 | 2 | · |
| **Nuts** | | | | | | | |
| Corylus avellana | Hazel | endocarp fragm. | 1 | · | · | · | · |
| Prunus dulcis | Almond | testa, complete | 8 | 35 | 15 | 17 | · |
| Prunus dulcis | Almond | testa, half or less | 29 | 126 | 91 | 25 | · |
| Prunus dulcis | Almond | seed fragm. | √ | √ | √ | √ | · |
| Prunus dulcis | Almond | endocarp fragm. | · | 8 | 6 | 1 | · |

concentrations of only stigmas. Flotation was unnecessary, and would have posed a risk of damaging the delicate strands.

Morphological identification of seeds, fruits, nutshells, flower parts, and leaves was undertaken using a microscope (x8–80), with comparisons made to relevant literature [24–26] and to a modern reference collection of seeds at the Department of Archaeology and Ancient History, Lund University. Since recovery, all plant remains have been stored in containers with water. The plant remains were kept in water during microscopic examination.

After identification, all species were quantified from individual samples. In some cases (clove and almond), different botanical parts of a species were identified; these parts were counted as separate examples. Complete and incomplete specimens were also distinguished. The approximate quantity of saffron was obtained by pouring the water sample containing the stigmas into a measuring beaker, allowing time for the stigmas to settle to the bottom of the beaker, and reading the volume. Presence of saffron stigmas in the four sediment samples (present in the hundreds) is indicated with ($\sqrt{}$), as it is unclear if the stigmas were *in situ* or were deposited in the samples due to excavation activity during which some stigmas dispersed into the water and might have settled on the seabed within the excavated unit. Presence of small endosperm (seed) fragments from almond, leaf parts and undetermined specimens are indicated in Tables 1 and S1 by presence ($\sqrt{}$). All analyzed plant material are held at the Department of Archaeology and Ancient History, Lund University.

## Results

### The exotic spices, and the total botanical assemblage

All recovered botanical remains were subfossil specimens. Identified economic plant species are presented in Table 1. The preservation state is noteworthy. Many plant remains feature fruit flesh and skin, still colored (Figs 3–9) and the saffron retains its distinctive aroma after 527 years submerged. Excluding the saffron stigmas which were quantified by volume (450 ml), a total of 3097 plant remains representing 40 species were identified from the sediment samples (full dataset S1 Table). They comprise 1 species of cereal, 1 species of oilseed, 4 species of fruits and vegetables, 8 species of spices, 2 species of nuts, 1 species of medicinal plant, and 23 species of wild taxa. The spices are numerically the most abundant plant remains in the assemblage, representing 86%, followed by nuts 12%. Crops, fruits, and vegetables make up a small portion of all the edible plants recovered thus far from the shipwreck. Wild taxa are represented by 2% in the plant assemblage, and these most likely entered the deposits over time as contamination from the coastal environment. In addition to the identified plant remains listed in S1 Table, sediment samples contained small plant debris from marine algae and remains of marine animals (shells, ligaments, perioctracum from mussels).

The archaeobotanical data discussed in this paper include only economic plant species typically used in food preparation or medicinal application. Wild taxa of the local flora fall outside the scope of this research, and therefore are excluded. Notes on some critical identifications of plants rarely found or previously unknown from archaeological deposits are discussed below.

*Crocus sativus* (Fig 3). The most important part of the saffron flower for human use is one of its female organs: the style, ending in three red-orange stigmas. Identification of saffron was based on the presence of these stigmas, also known as filaments, strands, or saffron threads. The recovered saffron stigmas do not consist of smaller pieces that have broken apart over time, or from handling by divers or during laboratory preparation and examination. Instead, the stigmas have a characteristic trumpet-like shape at the end of the pestle, indicating that the recovered strands are intact. Today and presumably historically, the price of saffron is determined by its quality. Grading criteria include the length of the threads. High-grade saffron is

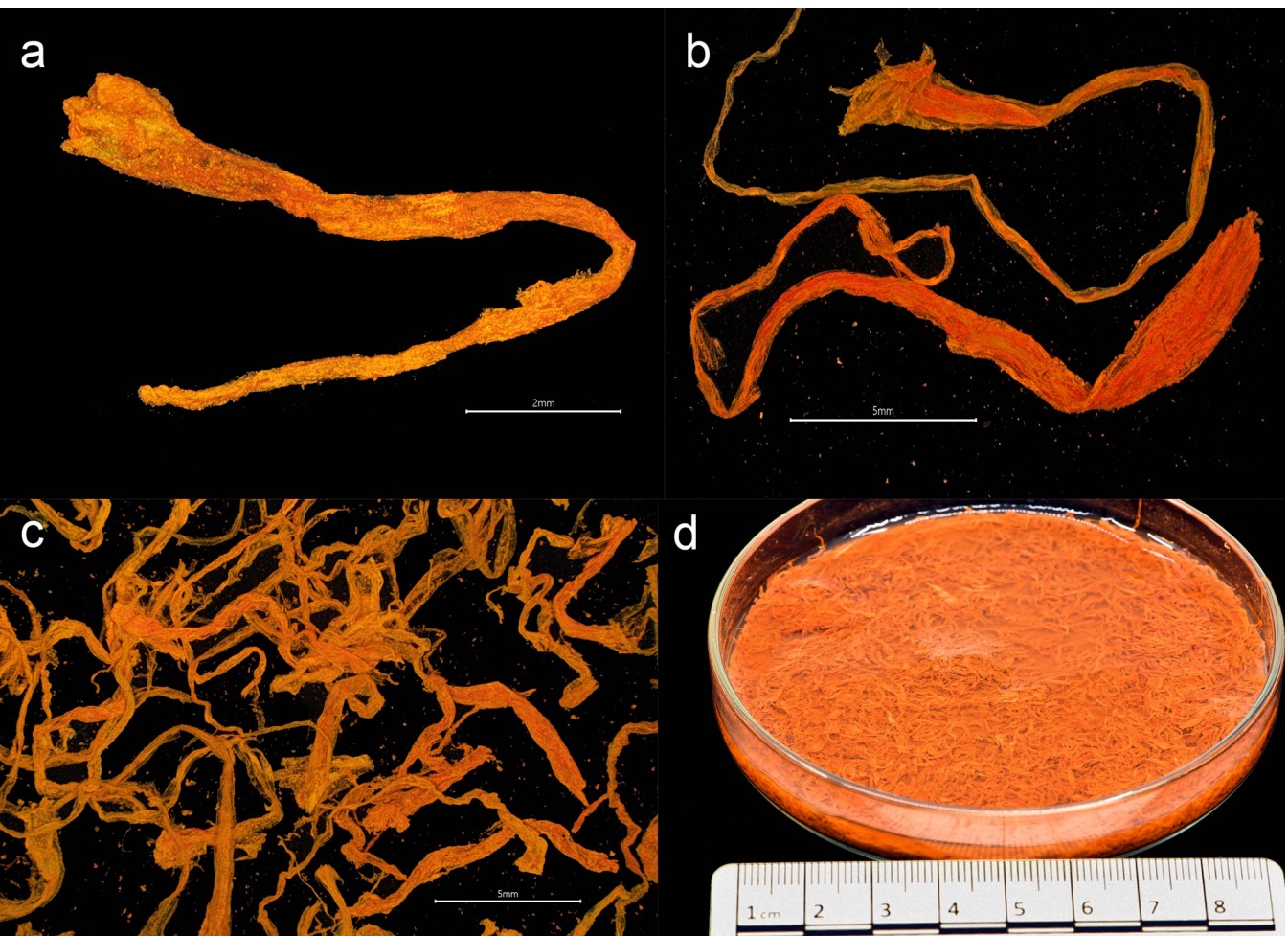

**Fig 3. Saffron from the *Gribshunden* shipwreck site.** Plant parts of saffron: a-c) stigmas, d) petri dish showing a portion of the recovered saffron stigmas.

composed of whole or cut threads (stigma with or without part of the style), and premium quality eliminates the yellow/orange part of the style [27]. Typical length of modern saffron stigmas are ca 2.5–3.5 cm [28]. Measurement of 50 *Gribshunden* stigmas with removed styles produced an average length of 2.36 cm. Thus, the *Gribshunden* saffron threads are consistent with modern grading for premium quality.

Several discrete concentrations of red material were noticed in sample 3, ranging in size from ca. 1x1 cm to ca. 5x5 cm. These were observed in the sample prior to wet sieving, and were removed from the sediment in which they were embedded. Each of the concentrations was placed in water, which quickly turned red/orange. Examination under a light microscope showed the concentrations were finely ground saffron stigmas; the water was colored as the powder dispersed into it. It is unclear how the ground saffron was stored on the ship, as no container or bag enclosing the powdery substance was observed. We hypothesize that the saffron might have been in textile bags which have not survived.

*Piper nigrum* (Fig 4). Black pepper was identified by the whole unprocessed fruits, in which the outer skin (epicarp) was complete or near complete among most of the specimens. The outer skin of the fruit shows the characteristic pattern of wrinkles from the drying process of unripe fruit. Some fruits entirely lacked their outer skin, or retained only fragments of it. This

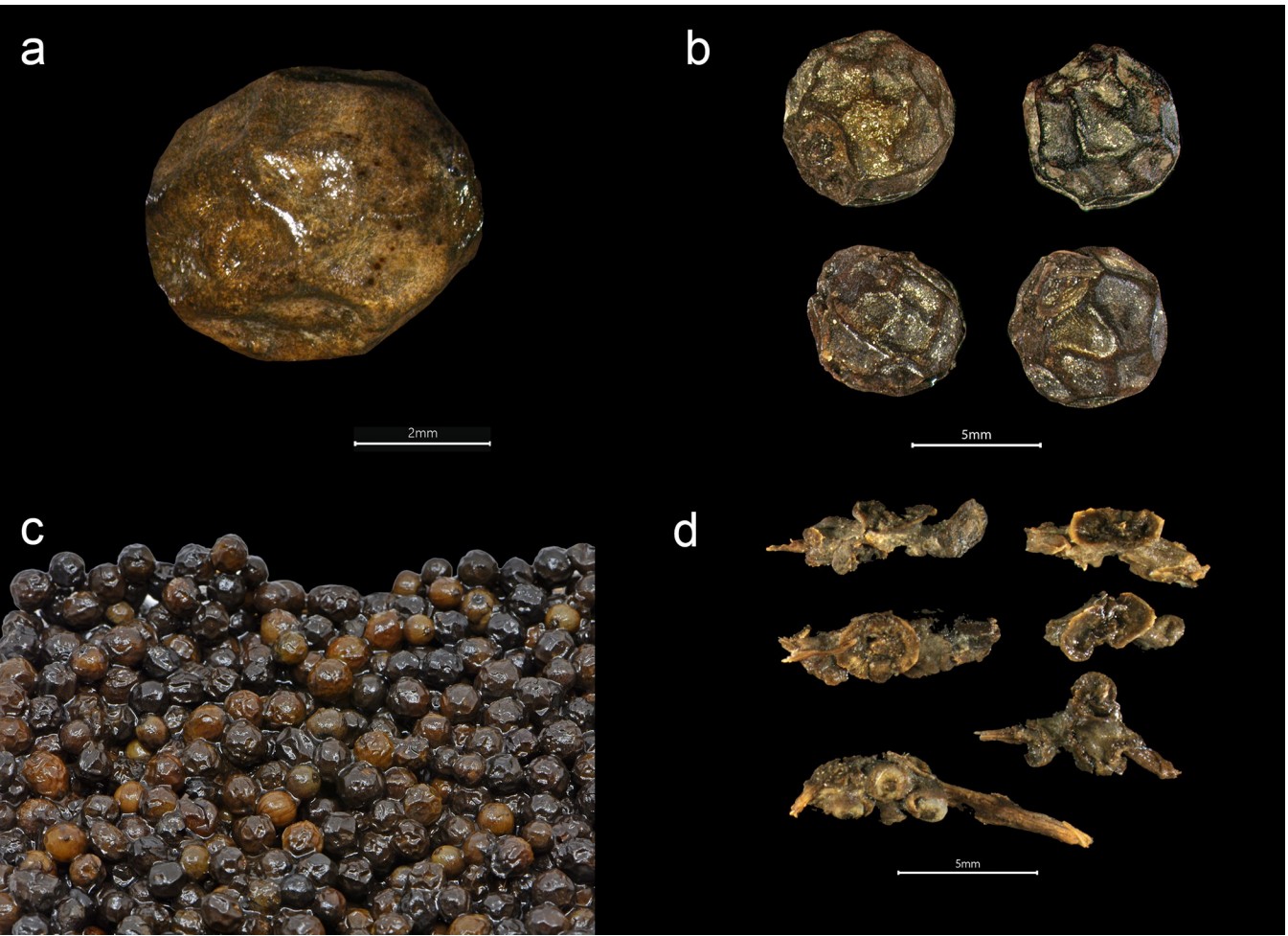

**Fig 4. Black pepper from the *Gribshunden* shipwreck.** Plant parts of black pepper: a-c) different views of peppercorns, d) stalk segments, some with unripe berries of pepper.

condition exposed the characteristic radial vessels of the fruit, resembling that of white peppercorn. Peppercorns with partially preserved and shriveled skin were able to be identified as black pepper. The loss of the outer skin surrounding the fruit is likely due to degradation over time or from handling during excavation or wet sieving. Some immature specimens and stalk parts were also recovered, including a few small stalk segments bearing unripe berries.

*Prunus dulcis* (Fig 5). Almond was identified by the presence of its seed coat (testa). While the seed coats of some specimens were complete, most were incompletely preserved with seed coat halves or mere shreds. Small white seed bits from endosperm were attached to some seed coats. Fragments of the nutshell (endocarp) from almond were also found among the samples, characterized by pitted canals containing vascular bundles.

*Syzygium aromaticum* (Figs 6 and 7). Cloves consist of a lower quadrangular stalked portion (the hypanthium) that terminates in four thick spreading sepals, which surrounds a globular head (the crown) having four imbricated petals. Identification of clove was possible from complete unopened flower buds preserved, but also from different fragmented parts of clove that have broken up over time. These parts include the hypanthium, the globular head, detached membranous petals that would have been attached to the globular head, and fragmented ovaries and ovules which have been released from broken up ovaries situated in the upper part of

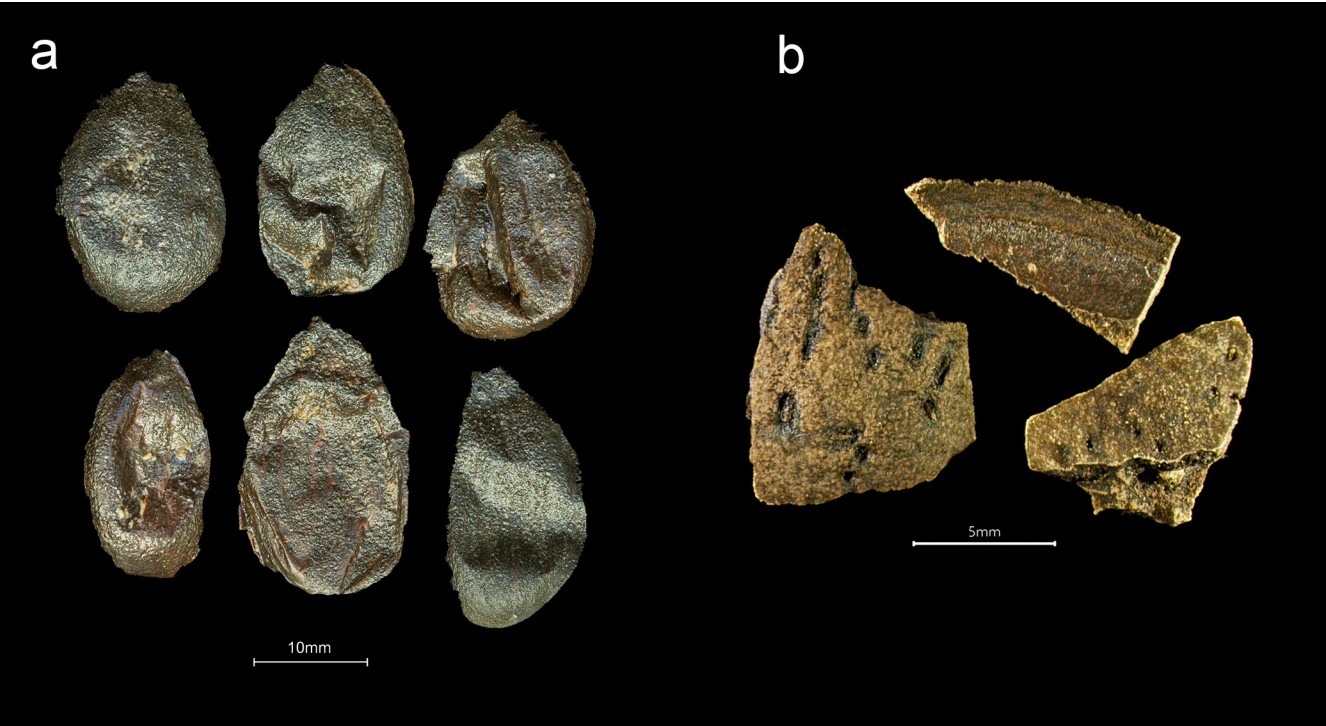

**Fig 5. Almond from the *Gribshunden* shipwreck.** Plant parts of almond: a) seed coats, b) nutshells.

the hypanthium. Clove stalk was also identified, as were numerous oil glands from the disintegrated hypanthium observed in the samples. Oil glands were not quantified, as these were present in the thousands.

*Zingiber officinale* (Fig 8). Ginger was represented by small parts of rhizome skin (epidermis) from the creeping underground stem. From rhizome skin parts, it was possible to identify the circular scars representing the nodes with small scales and buds, as well as small black dots on the surface of the skin. While most fragmentary skin was ca. 10–40 mm in length, a larger piece measured 60 mm. Adhering to some of the rhizome skins were whitish fibrous threads.

*Cucumis sativus* (Fig 9A). The seeds of cucumber can be difficult to distinguish from those of melon (*Cucumis melo)*, as the size of the seeds are variable. Identification was possible by examining the fine anatomy of the seed surface: *C. sativus* has fine straight parallel lines in the middle of the seeds' surface, and parallel lines which follow the oval shape of the seed. In comparison, *C. melo* have fine parallel lines of cells on the entire seed surface [29].

## Discussion

### Economic plants from the *Gribshunden* shipwreck and archaeological comparanda

The economic plant species from the *Gribshunden* botanical assemblage represent a wide range of food plants, presented here in five food categories: cereals, oilseeds, fruits and vegetables, spices and nuts. We acknowledge the overlap between culinary and medicinal applications of these plants. Henbane is the only non-food plant identified in this assemblage. Henbane had medicinal and magical purposes, and is presented under a separate heading. The archaeological context can help determine the most likely use for the recovered plants. All

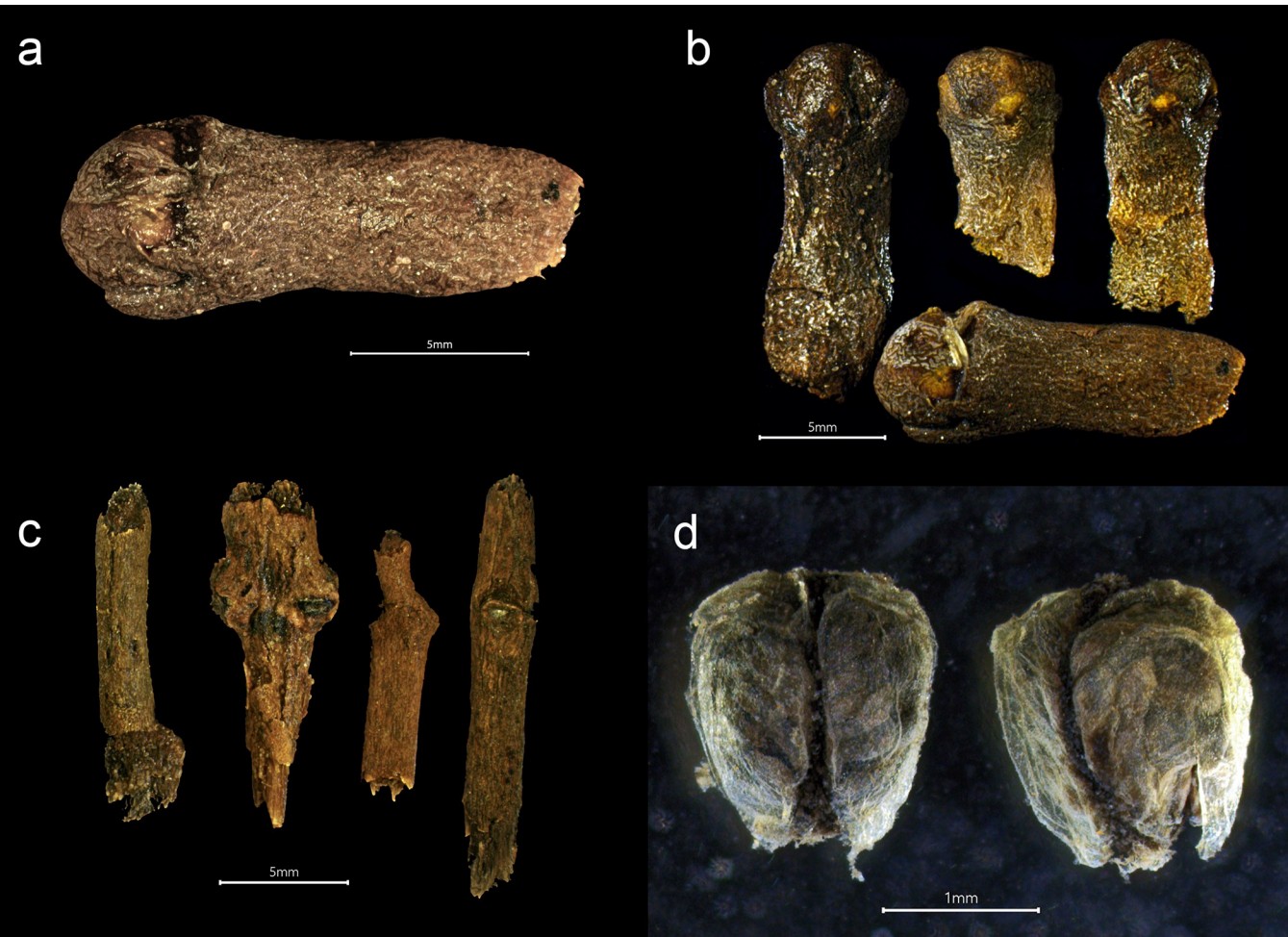

**Fig 6. Clove from the *Gribshunden* shipwreck.** Plant parts of clove: a-b) flower buds, c) stalks, d) side view of complete ovaries.

botanical remains from the *Gribshunden* shipwreck were recovered from sediments within a one square meter unit located toward the stern of the ship. The locus that produced the botanical remains also held several wooden casks presumed to have contained foodstuffs and beverages [3,23] Among these plant species, all are edible except henbane, which was used only medicinally. Then, as now, some of these plants are commonly consumed as flavorings (saffron, black pepper, clove, ginger, black mustard, dill, caraway), others eaten as snacks or used in baking (nuts, berries, grapes, flax), and some could be eaten as part of a meal (cucumber). Grapes could have been consumed as raisins. Raisins and almonds could be consumed alone or as ingredients in prepared dishes.

It is unclear how these food plants were stored and transported. No containers of wood, metal, glass, or ceramic were excavated in the same locus. Similarly, this locus produced no remains of bags made from textile or leather, though it is possible that bags simply have not survived. Shells from nuts and ground saffron (conglomerates of ground stigmas) were recovered in the botanical assemblage, suggesting that these foods were handled and prepared close to the sample area shortly before the ship sank. From this, we propose that the economic plant remains were foremost intended as foodstuffs.

We speculate that the ground saffron was separated into several small packages. This possibly suggests medicinal doses or discrete volumes of flavorings for food or beverage servings; it

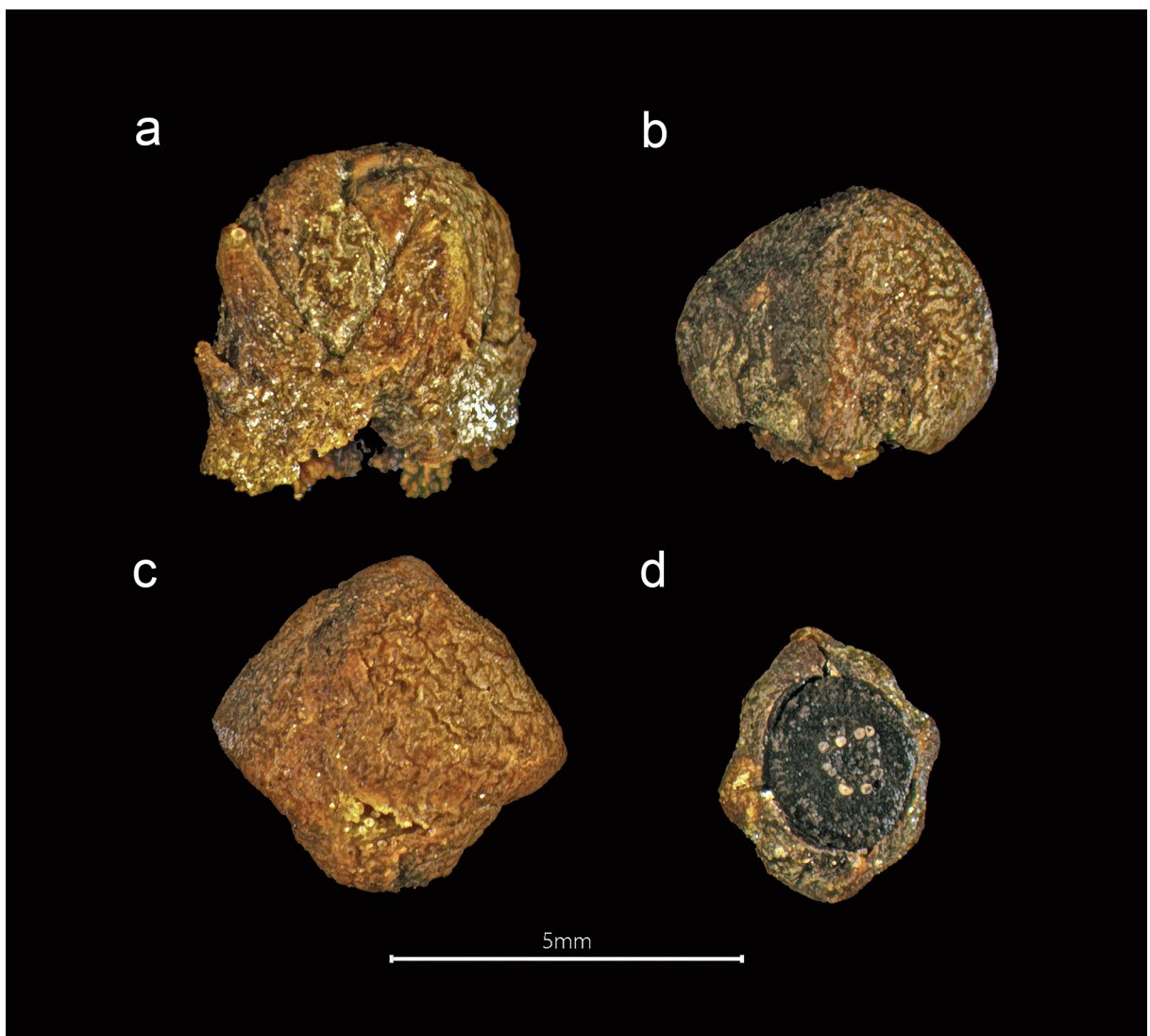

**Fig 7. Clove from the *Gribshunden* shipwreck.** Different views of detached globular head of clove: a) side view with attached petals, b) side view without petals, c) proximal side (above) of globular head, d) distal side (below) of globular head.

could also indicate division based on ownership, or even packaging as gifts. However, detailed discussion of specific culinary use or healing properties of respective food plants during the Middle Ages is outside the scope of this paper. Medicinal preparation and application of these spices in medieval Europe was diverse, and information from Scandinavia is at best fragmentary. An overview of previous archaeobotanical data from the medieval Baltic Sea region with reference to selected food plants recovered from *Gribshunden* is summarized in Table 2. For each of the five food categories, we present the individual *Gribshunden* plant species with a summary of relevant published archaeobotanical documentation for medieval Scandinavia and the Baltic region. The archaeological record is, of course, incomplete. Cereal grains, oilseeds, nutshells, or aromatic fruit or seed condiments are sometimes represented in archaeological deposits. However, those plants used for their vegetative parts rarely survive archaeologically, unless preserved in the waterlogged contexts of latrines, wells, or underwater environments. Preservation of food plants in the macrofossil record may further be prevented

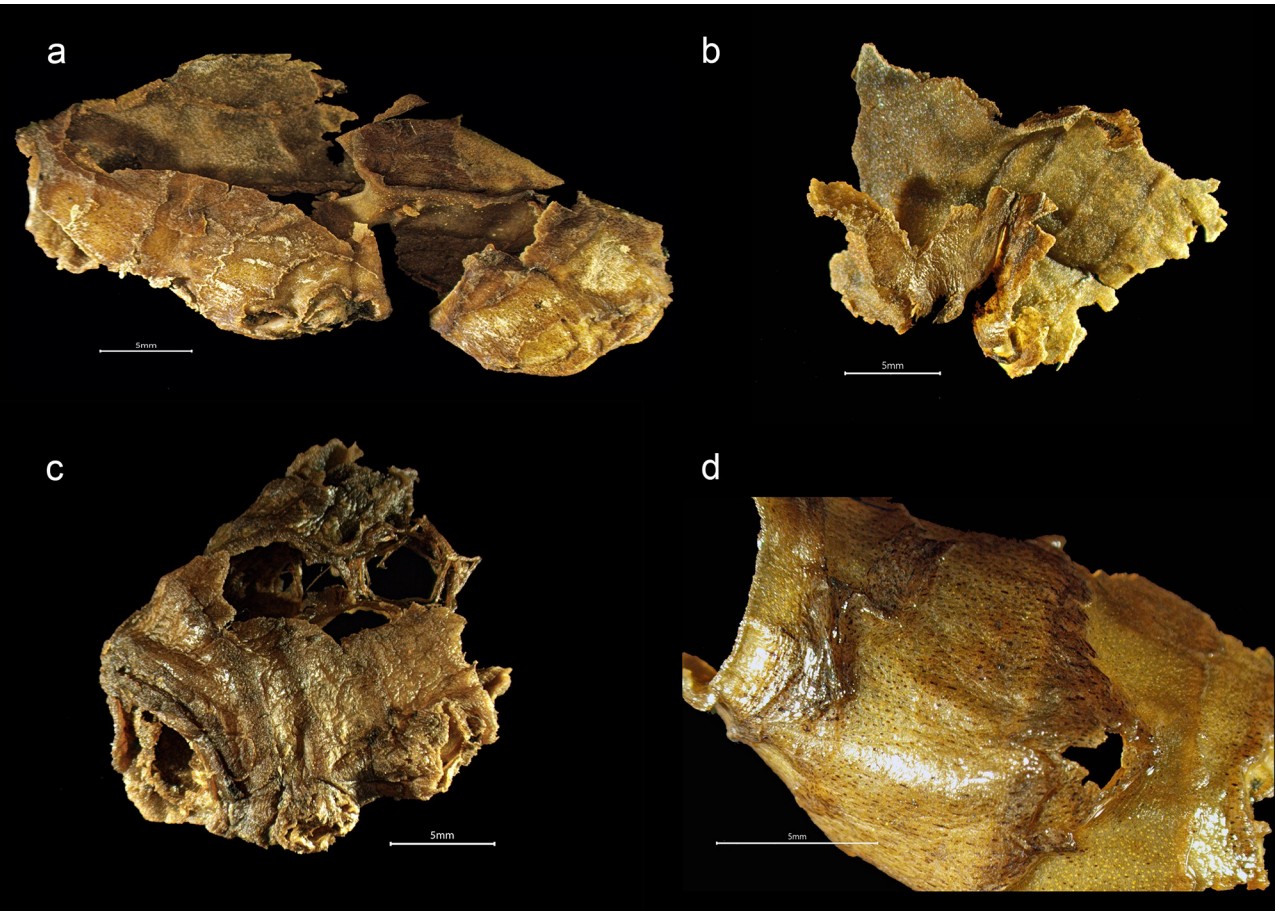

**Fig 8. Ginger from the *Gribshunden* shipwreck.** Plant parts of ginger: a-c) rhizome skin showing scales and auxiliary buds, d) close-up of black-spotted surface of skin.

if processed and used in other forms, such as ground or as powder, or pressed into oil. Written sources that mention rare food plants can, however, complement archaeobotanical data with insights into the procurement and consumption of these plants.

**Cereals.** Cereal grain was present in small quantities of a few seed coats from wheat grains (testa), with one grain probably bread wheat, also known as common wheat. Whole unprocessed cereal grain found on shipwrecks is often cargo. However, *Gribshunden* was a warship on a diplomatic mission, not a cargo-laden trading vessel. It is unknown if *Gribshunden* carried unprocessed grain as provisions, as most of the hold has not yet been investigated. A 1493 provisions list of another Danish warship, *David*, includes flour and bread [38]. This ship was equipped with a bakery and full galley. Similar facilities are not mentioned in accounts concerning *Gribshunden*. Considering the substantial time the king spent aboard his flagship, it must have been similarly outfitted; future excavations are likely to reveal these features. While it is likely that much of the bread and biscuit consumed by the common sailors was supplied from shore, it is probable that a bakery on board *Gribshunden* provided fresh baked goods for the noble passengers. The recovered wheat grains could have originated in these supplies of wheat flour, or as contamination among other products.

Until modern times, bread wheat was considered a luxurious cereal in the Baltic region. Used for making cakes and bread, it was appreciated for its white flour and dough properties.

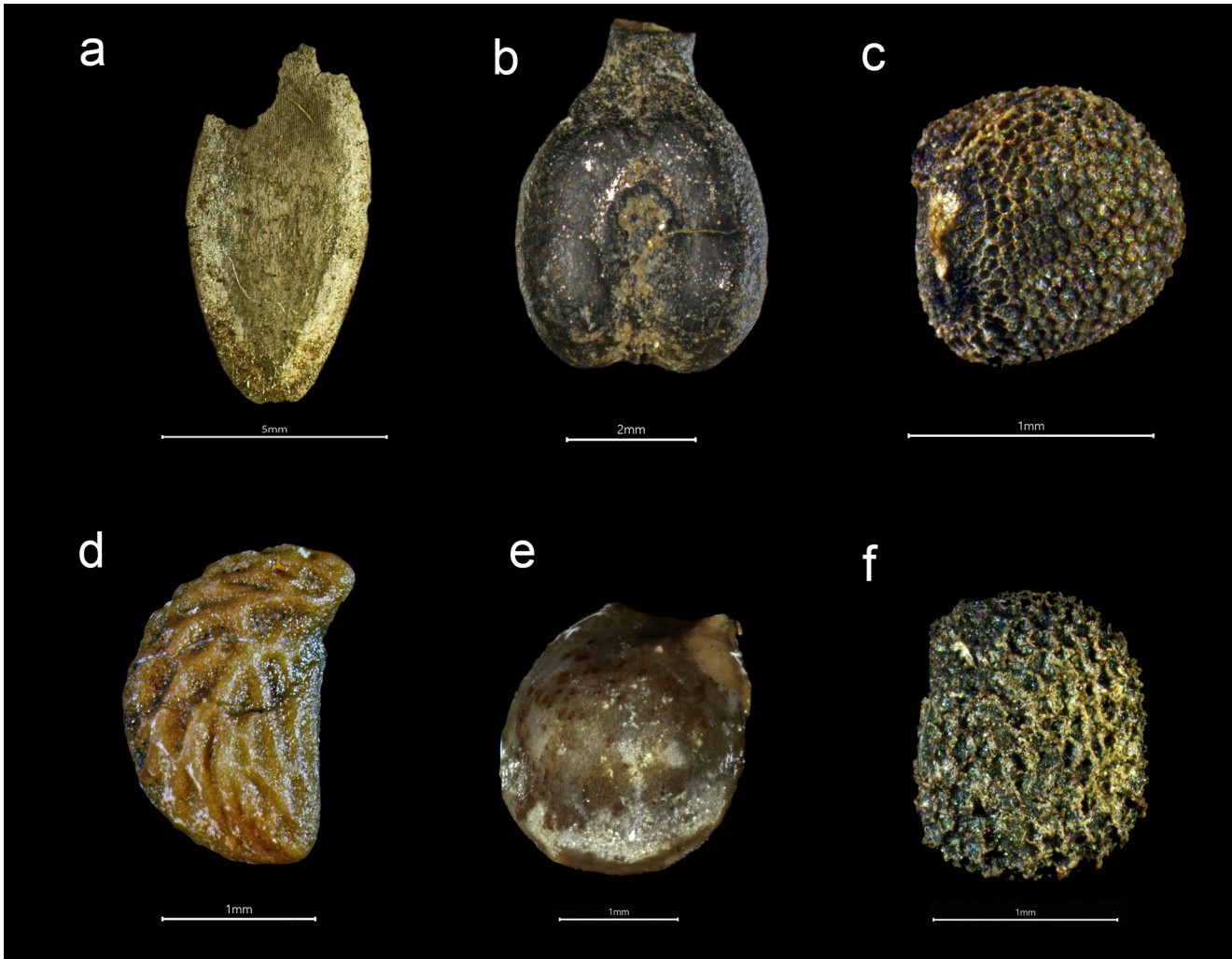

**Fig 9. Variety of plant species from the *Gribshunden* shipwreck.** Seeds of: a) cucumber, b) grape, c) black mustard, d) raspberry, e) hop, f) henbane.

It produced a leavened loaf not possible to make with flour from the other wheat species [39]. Bread wheat's medieval exclusivity might lie in dietary traditions and fashions, but it might also be due to its cultivation requirements. It is the most demanding species of all cereals, requiring humus-rich soils and laborious crop processing regimes. From 15th century Scandinavian accounts, cultivation of wheat is described as problematic as it was difficult to keep seed grain pure; rye would start to grow among the wheat rows, and after a few years it dominate the field [40]. Rye and barley were the more common cereal types in medieval Scandinavia, followed by oats [35,41]. Rye was used for bread making. Barley sometimes was used as food in whole-grained groats, but its primary use was in beer production. Oats were sustenance for both people and their horses.

**Oilseeds.** Plants with oil-rich seeds are represented on *Gribshunden* by flax. This plant has played an important economic role in Northern Europe since its introduction in the beginning of the first millennium CE. It is commonly found in medieval Baltic domestic archaeological contexts [42,43]. The plant was valued for its nutritious seeds, consumed whole by adding them to foods or pressed to extract oil, or used in medicinal preparations. Besides its seeds,

**Table 2. Previously recorded archaeobotanical finds of imported or rare food plant found at the *Gribshunden* shipwreck that illustrates the evidence of these in the Baltic Sea region during a time sequence from the Early to Late Medieval Age using presence and absence.**

| Timeline (CE) | | Plant species | Almond (*Prunus dulcis*) | Black pepper (*Piper nigrum*) | Clove (*Syzygium aromaticum*) | Cucumber (*Cucumis sativus*) | Ginger (*Zingiber officinale*) | Grape (*Vitis vinifera*) | Saffron (*Crocusa sativus*) |
|---|---|---|---|---|---|---|---|---|---|
| | | Countries | | | | | | | |
| Early Medieval Age (1050–1300) | 1100 | Denmark | | | | | | | |
| | | Sweden | | | | | | | |
| | | Northern Germany | | | | | | • | |
| | | Northern Poland | | | | | | • | |
| | | Estonia | | | | | | | |
| | | Finland | | | | | | | |
| | | Norway | | | | | | | |
| | 1200 | Denmark | | | | | | | |
| | | Sweden | | | | | | | |
| | | Northern Germany | • | • | | | | • | |
| | | Northern Poland | | | | • | | • | |
| | | Estonia | | • | | | | • | |
| | | Finland | | • | | | | | |
| | | Norway | | | | | | | |
| Late Medival Age (1300–1550) | 1300 | Denmark | | | | | | • | |
| | | Sweden | | | | | | • | |
| | | Northern Germany | | • | | | | • | |
| | | Northern Poland | | • | | • | | • | |
| | | Estonia | | | | | | • | |
| | | Finland | | | | | | | |
| | | Norway | | | | | | • | |
| | 1400 | Denmark | | | | | | | |
| | | Sweden | | | | | | | |
| | | Northern Germany | | • | | | | • | |
| | | Northern Poland | | • | | | | • | |
| | | Estonia | | • | | • | | • | |
| | | Finland | | | | | | • | |
| | | Norway | | | | | | | |
| | 1500 | Denmark | | | | | | | |
| | | Sweden | | | | | | | |
| | | Northern Germany | | • | | • | | • | |
| | | Northern Poland | | • | | • | | • | |
| | | Estonia | | | | | | • | |
| | | Finland | | | | • | | • | |
| | | Norway | | | | | | | |

Species that are commonly represented in medieval contexts are not included (i.e. wheat, flax, hop, local spices, hazel, berries, and henbane). The archaeobotanical records are based on [12,30–37].

flax was an important source of raw material in fiber production. The widespread documentation of the seeds in archaeological layers indicates that the plant was commonly used across all strata of society. The occurrence of flaxseeds on *Gribshunden* indicates consumption of the whole seeds, perhaps added to foods such as breads and porridge.

**Fruits and vegetables.** The plants in this category consist of seeds from four taxa: cucumber, grape, raspberry and blackberry. Cucumber's botanical classification is as a fruit; but because it is often eaten as part of a main meal, its culinary definition is as a vegetable. Grape is a fruit, consumed fresh or dried as raisins, and in the medieval era it often was a dessert ingredient on elite tables [11]. Raspberry and blackberries are aggregate fruits, commonly referred to as berries, and these are indigenous to woodland areas across the Baltic region. Fresh berries are highly perishable, with shelf lives extending only a few days. However, preservation by drying or other means could prolong their storage life. The seeds from these berries occur frequently in medieval archaeobotanical material from both urban and rural contexts.

Archaeobotanical remains of cucumber from the Baltic region are rare for any period [35,44]. The few reported finds come from urban cultural layers of latrines or backyard deposits. A single seed dated to the 13th century has been recorded from a latrine from the coastal Hanseatic city of Gdansk, Poland, and cucumber seeds dated to the 8th-12th centuries have been reported from the Polish interior [45]. In Tartu, Estonia, cucumber seeds have also been found in excavations of 15th century latrines [46]. By the 16th century, seeds of cucumber become more frequent in archaeological layers from towns along the southern Baltic coast [35], and historical documents from Gdansk mention that cucumbers were grown in gardens [45]. The only medieval botanical evidence of cucumber in Scandinavia is a late medieval find of pollen from a latrine at Svendborg on the island Fyn, Denmark [47].

Grape seeds, or pips, dated to the medieval period are infrequent in Scandinavian archaeology. Grape has been reported from Bergen, Norway dating to 13th-14th century [48], and a few single pips dated to the 14th century have been reported from Sweden and Denmark [35,49]. Grape is more commonly encountered in excavations of Hanseatic towns in northern Poland, northern Germany, and Estonia [12,45,46,50].

If the primary consumption of grape products was as wine, and raisins did not play a significant role as imported goods to Scandinavia, this may explain the scant archaeobotanical finds of grape pips. While most grapes likely were imported from more southern regions, some level of local Baltic cultivation might have been attempted. From the 17th century, a written account from Poland reports that grapes did not ripen and were of poor quality [37]. A Danish bishop mentions in his diaries how the September frosts in 1694 and 1695 caused the local grapevines to lose their leaves [51]. Before the Middle Ages, sporadic finds of grape pips dating to 800–1000 CE have been documented in southern Scandinavia: at the trading town of Hedeby in Northern Schleswig [52], in a Viking Age grave on Gotland, Sweden [53], and from a royal Viking Age complex at Tissø, Denmark [54].

**Spices.** The eight species listed in this category are commonly employed as food flavorings, using the plants' seeds, flowers, or rhizomes. Of these, saffron, black pepper, clove, and ginger are from geographical origins far distant from Scandinavia. We refer to these four as exotic spices. The others are species that grow well in the Baltic region: black mustard, dill, caraway, and hop. We refer to them as local spices. Beyond their properties for flavoring and coloring food (saffron), they were also used as food preservatives and medicine [55]. Their presence on *Gribshunden* complements the Scandinavian and the Baltic region archaeobotanical record, and their mentions in medieval written sources.

*Exotic spices*. The exotic spices identified from *Gribshunden* were luxurious foods in medieval North Europe, imported through a chain of traders extending to Asia. These are represented in the *Gribshunden* botanical assemblage by fruits (black pepper), stigmas (saffron),

flower buds and stalks (clove) and rhizome skin (ginger). These plant parts were abundant in the samples from the wreck; this is remarkable because of their archaeological rarity. The delicate fragments of these plants seldom survive in terrestrial archaeological contexts. Spices from Asia were introduced to western and central Europe by the Roman legions during the first millennium CE [56,57], but archaeobotanical material and written sources show that such exotic spices first reach Northern Europe only in the early medieval period [58].

After the development of the Hanseatic League in 12[th] century, German merchants dominated Baltic trade networks, displacing Scandinavians traders. Historical documents indicate that the Hanse traded exotic spices in the Nordic region, conveyed from the Mediterranean and Asia through the continent to the Baltic coasts [58]. Exotic spices were known in medieval Scandinavia, as evidenced by a handful of historical sources. An early 13[th] century French cookbook was copied in courts across Europe, and translated to Danish by Henrik Harpestreng (d. 1244) [59,60]. It lists thirty-one abbreviated recipes, with dishes including ingredients and spices available only to the highest social strata: pepper, saffron, clove and ginger. The 1231 cadaster of the Danish King Valdemar II mentions one exotic spice, pepper, in a list of provisions that the king and his court expected on visitations within his kingdom [61]. A 1315 bequest to the abbey of Sorø, on the island of Zealand, Denmark, records that one course of a meal for duke Christopher of Halland and Samsø (he later became King Christopher II), was to be made with pepper [62]. In Sweden, according to an account from 1328 of a merchant in Stockholm, saffron, pepper and ginger were among the spices sold to the organizers of the burial of Birger Persson, the father of Saint Birgitta [63,64]. Later, an estate inventory from 1365 of Queen Blanche of Norway and Sweden lists clove among her spices [65], and in Stockholm in 1467, an account from a religious order mentions pepper and saffron [66].

Medieval written sources from the southern Baltic, especially from the Hanseatic towns in areas of northern Germany, northern Poland, and Estonia, reveal that a greater variety of exotic spices was more frequently consumed there than in Scandinavia, and by a wider but still exclusive segment of the population [12,45,46]. Studies of inventories and purchase lists from Baltic merchants, city councils, guilds, and wealthy households indicate saffron, pepper, clove and ginger were consumed by an expanding elite [11,12,58].

Previous archaeobotanical data from the Baltic region concerning the exotic spices found on *Gribshunden* are summarized below.

Black pepper is indigenous to the western coast of South India [67,68]. The peppercorns can be used ground, dried, or added whole to foods. While black pepper is mostly employed in cooking as a food seasoning or as a table spice, peppercorns also has medicinal uses. The spice could also have been used to make rancid meat palatable while at sea [69].

Among the four exotic plant species found on *Gribshunden*, black pepper is the only spice previously found in the Baltic region archaeobotanical record. The early finds of black pepper are rare and consist of a few fruits, an.d these are mostly documented in excavations in Hanseatic towns along the southern Baltic coasts. They occur first in Bremen, Rostock, and Oldenburg, Germany in the 13[th]-14[th] century [70]; then somewhat later in Gdansk and Elblag, Poland [71]; and Tartu, Estonia in the 14[th]-15[th] century [43]. Its rare archaeological appearances are only in contexts of exclusive buildings associated with high-ranking individuals, suggests pepper consumption was limited to the social elites during this period.

Archaeological documentation of peppercorn in medieval Scandinavia occurs at two sites: a single peppercorn from the 13[th] century cathedral in Turku, Finland [44], and from a 15[th] century latrine in Næstved, Denmark [72]. While finds of peppercorns increase somewhat in 17[th] century archaeological contexts, they are still rare. Typically finds are from latrine and waste deposits in urban contexts, such as in Copenhagen [73], and Stockholm and Jönköping, Sweden [74,75]. The Danish Bishop's hall at St. Botolph's church produced a single peppercorn

[51]. These latter finds coincide with the establishment of several charter companies in northern Europe during the 17th century, including several East Indian Companies, which triggered an expansion of spice importation to the Baltic region.

Saffron joins the other three species previously not found in the archaeological material from Scandinavia or the Baltic region. Saffron is made from the dried stigmas of *Crocus sativus*, the saffron crocus. It is highly valued for flavoring foods and for coloring them golden-yellow, and for its medicinal properties. The delicate stigmas typically defy archaeological preservation. The recovery of an abundance of saffron stigmas and lumps of ground saffron from *Gribshunden* is exceptional; no archaeobotanical remains of saffron have previously been reported.

The geographical origin for saffron is not entirely understood. It is believed to have originated in the eastern Mediterranean, and was grown in Southwest Asia and the Mediterranean basin in Classical times [76]. The earliest evidence of saffron comes from iconography and artistic floral motifs of the Bronze Age Minoan culture in the Mediterranean. The ceramic Kamares Cup from Knossos holds an early iconographic representation of the crocus flower dating to 2100–1800 BCE [77]. A fresco depicting saffron in the Palace of Knossos in Crete dates to ca 1700–1600 BCE [78]. Another fresco, dated about 1500 BCE, is the 'Saffron Gatherers' at Akrotiri on the island of Santorini [79,80]. Plant ideograms found on Mycenaean Linear B tablets (1400–1100 BCE) perhaps indicate saffron [81,82]. Cultivation of saffron spread to Spain no later than the 10th century CE, and by the 14th century CE, Spain was a prominent exporter of saffron [83]. While the medieval Baltic trade in exotic spices expanded at the end of the 15th century, the geographic origin of the *Gribshunden* saffron is as yet undetermined; future aDNA analysis may suggest its source.

Clove and ginger are the two other *Gribshunden* exotic spices that are without precedent in the medieval Baltic macrofossil record. The aromatic flower buds of clove are commonly used as a spice or for medicinal purposes. The rhizome of ginger can be used fresh, dried as a powder, or preserved in vinegar for similar purposes. The origin of ginger is not known, but it is believed to have evolved in Southeast Asia [84]. Cloves are native to Indonesia [85]. Trade in these spices from India to Europe can be traced back at least as far as the Classical period [75,86]. These spices probably first reached the Baltic region in the Middle Ages through Hanse middlemen [35].

Archaeologically, finds of clove and ginger are both rare and geographically sporadic. The only possible clove find in the Baltic region is pollen detected in a 17th century latrine in Copenhagen [47]. It was tentatively identified as Myrtaceae, and interpreted as clove. Beyond the Baltic, clove pollen has been documented in Dutch cesspits dating from the 16th to 19th centuries [87]. Macrofossil finds of clove have been reported from a few sites. At Les Jacobins convent, France, a few fragments of the flower bud, dated to the 17th century, was documented among embalming plants [88]. At the ancient port of Mantai, Sir Lanka, a site related to the early days of Indian Ocean trade, one clove was found dating to ca. 900–1100 CE [89]. Finds of ginger are limited to two archaeological sites: dry roots reported among foodstuffs in a tomb of a Han woman who died in China 168 BCE [90], and from an excavation at the Port of Qusier al-Qadim, Egypt ca. 1050–1190 CE [91].

*Local spices*. Spices in this group are plant species that grows well in Scandinavia and were commonly available in the medieval period. These food and beverage flavorings are connected to garden cultivation, and include black mustard, dill, caraway, and hop. Hop is typically not defined as a spice, but we include it in this category because of its primary use as a flavoring and stability agent in beer. Dill and caraway can both be considered spices or herbs depending on which plant parts are used. Their seeds are frequently used to spice stews and soups, or to

flavor baked goods. When the aromatic leaves are utilized in cooking or eaten raw, they are commonly described as herbs; or if the root is consumed, a vegetable.

The seeds, aromatic leaves, and roots of the plants in this group are all used for cooking. While all of these parts could be preserved on the shipwreck, only their seeds have been recovered from *Gribshunden*. Black mustard was most plentiful in the samples, while the other species are more sparsely represented. The seeds of black mustard can be pressed to produce oil, but the predominant use is as a spice and in the production of mustard. It is likely that black mustard, dill, and caraway were available at hand for day-to-day culinary use during the voyage.

Archaeobotanical records from Northern Europe show that some level of gardening existed parallel to arable cultivation beginning in the Early Iron Age [92]. Black mustard and dill are among the spices utilized earliest. They likely were introduced from northern areas of continental Europe, where the Roman occupation influenced food production in Germania and spread the knowledge of garden cultivation [36]. Sporadic finds of black mustard and dill seeds in archaeological contexts are reported from the last centuries BCE and the first centuries CE in southern Scandinavia [93,94]. The diversity of seeds from plants linked to gardening increases in Northern Europe around 800–1000 CE [32,92]. During this time, caraway and hop are occasionally recorded in archaeological layers.

An expanding diversity of species in early medieval Scandinavia coincides with the establishment of self-sufficient monasteries producing vegetables, fruits, herbs and spices at the beginning of the second millennium CE. In the developing urban centers of the 11[th] and 12[th] centuries, the archaeobotanical record indicates that horticulture became an important part of household autonomy [31,95,96]. Seeds from black mustard, dill, caraway and hops are represented in many medieval contexts, particularly in urban cultural layers [31,36,97,98]. As such, these species seems to have been well-integrated into local medieval Scandinavia cultivation. Similarly, these species are frequently encountered in archaeobotanical material from medieval southern Baltic coastal towns [58]. The widespread presence of seeds from black mustard, dill, caraway and hops in medieval archaeological layers around the Baltic region suggests that these were commonly grown in gardens, and it is therefore possible that local producers supplied these plants to *Gribshunden*. Nevertheless, produce that could be supplied locally may still have been sourced from long-distance networks. A study of the Swedish Navy in the 16[th] and early 17[th] centuries describes procurement of staple food supplies such as meats, grain, peas, butter, and salt through networks of Baltic ports, but there is no mention of sourcing spices. In that era, vegetables, bread, and beer were typically sourced in proximity to shipyards or picked up during the journey. During the first leg of a voyage it was not uncommon that each sailor brought his own food [99].

Hop was represented in the botanical assemblage by a single fruit (seed). Unlike the other local spices, hop seeds are not utilized. For this reason, only female plants are grown in hop fields to prevent the ripening of fruits. Male plants are cultivated separately for breeding purposes. The flavor and aroma of hops are instead from the oils of the hop cone. Hop's foremost use is as a flavoring and stability agent in beer-making, and the frequency of its archaeological occurrence may be a testament to the vital role of brewing in medieval society. The oils contained in the cones possess antibiotic properties important to the brewing process. Hops suppress bacterial growth, while allowing brewer's yeast to thrive. Hop fruits are common in medieval archaeobotanical material from the Baltic region. However, the seeds found in terrestrial archaeological layers are likely a result of unintentional pollination and dispersal of seeds when handling hop cones [100,101]. The single hop seed recovered from *Gribshunden* suggest that it was accidental contamination, possibly coming in with the beer in barrels, or among other foodstuffs.

Information about local horticulture from Scandinavian written sources of the 15th century is scarce, with the exception of hop, as the late medieval period witnessed the beginnings of domestic brewing regulation. Scandinavian laws from 13th-15th century stipulated farmers' obligation to grow certain quantities of hops, and also regulated sales of hops [102,103]. Hop is frequently mentioned in Hanseatic towns' pile duty books (documents recording fees paid by incoming and outgoing vessels) as a plant produced in specific areas [12]. Cultivation regions included Zeeland in the Netherlands, the coast of Jutland, Denmark, and areas of contemporary Poland. From there, it was exported to urban areas around the Baltic for use in commercial and domestic breweries [12], indicating that beer and malt were commonly traded throughout medieval societies [10,104,105].

**Nuts.** This category comprises two species of nuts: almond, which originates from southern regions with milder climates and was imported to Scandinavia; and hazel, which is indigenous to the Baltic region, but might also have been cultivated. Nuts can easily be stored for long periods. They are an oil-rich food high in calories, and an important source of protein and carbohydrates. They could be eaten raw on their own, added to main dishes, or used in desserts.

Almond was mainly represented in the samples by the seed coat, but also by some shell fragments and bits of endosperm still attached to some seed coats. Archaeological finds of almond are exceptionally rare from the Middle Ages in the Baltic region. The earliest evidence is from Oldenburg, Germany, dated to the 13th and 14th centuries [50], while a later find dates to 17th century Poland [45]. Archaeological excavations conducted in 1937 at Aalborg, Denmark, produced almond shells dated to ca. 1700 [51]. From medieval Scandinavia, evidence of almond consumption is limited to a few written references to the nut: the 1467 account from a monastery in Stockholm [66], and a 1578 post-medieval account of spices purchased for the Swedish royal court [106]. Studies of archival documents associated with some festival purchases in 15th and 16th century Reval (modern Tallinn, Estonia) and Riga, Latvia indicate the upper classes consumed luxuries including almonds during festivals [11]. Almond was particularly versatile on royal tables, used in spiced almond milk, or even in marchpane (marzipan) [107,108].

In contrast to almond, shells from hazelnut are common in both urban and rural medieval contexts around the Baltic region. Evidence of its use dates back to the Mesolithic [109–111]. Hazel is native to Europe and grows wild as far north as central Scandinavia. But it also has been cultivated, as management of hazel forests is mentioned in medieval Scandinavian legislation [37]. The species was valued primarily for its nuts, but its leaves were also used for animal fodder [112].

**Medicinal plants.** Henbane is a poisonous plant often connected to cult, magic, and medicine, e.g. as drug, poison, and in amulets [113]. In small doses henbane could be used as a general analgesic. In large doses it could be used as an ingredient in witches' ointment. It also acted as a love philter, and at highest doses it was a poison [114–116]. It is indigenous to the Mediterranean region, but was introduced to Northern Europe during the Pre-Roman Iron Age, about 400–300 BCE [92,117]. From the first millennium CE henbane seeds occur occasionally in Scandinavian contexts [118]. Henbane is widely detected in various urban medieval contexts throughout northern Europe, likely due to cultivation and common use [119]. However, the plant easily spreads from gardens to other areas around living quarters, so henbane also may represent a weed in urban layers [120,121].

Descriptions of early medieval Scandinavian medicinal use of henbane were recorded by Henrik Harpestreng, physician to Danish King Erik IV. In his herbals *Liber Herbarum* [122], he recommended direct application of the seeds to relieve toothache. This palliative is also described in a 15th century Swedish medicinal manuscript [123]. Other prescriptions directed heating the seeds over a fire or a hot iron, and leading the resulting smoke onto the aching

tooth [114]. Similar treatment was described in other parts of Europe, for example by the herbalist of Queen Elizabeth I in the late 16[th] century [116].

The henbane seed from *Gribshunden* may represent onboard medicinal supplies, carried for analgesic purposes. Alternatively, with only a single seed found on *Gribshunden* so far, it may have been accidental contamination among other food supplies. Given the range of wild taxa in the botanical assemblage, another possibility is that the henbane originated from coastal vegetation. If future excavation on the wreck produces greater numbers of henbane seeds, its purpose may become clear.

Medicinal benefits are possible for the other spices. Then, as now, various food plants were present both in the kitchen and the apothecary. For example, saffron, pepper, cloves, and ginger could have been consumed both for their aromatic and medicinal properties when added to food or beverages. Contextually, the range of economic plant species from *Gribshunden* represent those typically used in food preparation; only henbane is a non-food plant. For this reason, it is possible that these spices were foremost linked to food consumption, rather than for medicinal use alone.

One exception may be hop. Outside of beer-making, hop also has medicinal application and may have been brought onboard for that use. A sedative effect of hop is its most common medicinal application, able to induce sleep [124]. Bedclothes could be stuffed with plants believed to bring calm: for instance, the fresh odor of hop cones, together with lavender and lemon balm. A Scandinavian archaeological example exists from the casket of a Swedish bishop buried in Lund in 1679 [125]. The stuffing of the pillow and mattresses upon which he laid were filled with plants including hop cones and an abundance of hop seeds. No hop cones have yet been found in the botanical assemblage from *Gribshunden*; interpretation of similar use is speculative.

## Elite consumers of plant foods recovered from *Gribshunden*

Knowledge about historical victualling and the diet of ships' crews is primarily derived from written sources. Information on maritime foodstuffs is scant for 15[th] century Scandinavia, and in the case of *Gribshunden*, no records detail the diet on board during its last voyage. For the years 1487 and 1493, however, documents provide some expenses for King Hans' naval fleet [38]. The documents are scattered summaries kept by the king's secretaries, usually recording supplies of arms and recruitment of soldiers. Notes from 1493 mention funds set aside for ships' crew victuals. They consumed meat, butter, fish, bread, flour, salt, and vinegar. The accounts relate an ordinary and monotonous diet, similar to the fare of other sailors in medieval northern Europe. Provisions aboard three 16[th] century Swedish naval ships (*Lindormen* 1546, *Sankt Erik* 1561, *Vita Falken* 1562) list the same foods, with the addition of peas [126]. Nowhere is there mention of spices, vegetables, fruit or berries for the common Jack Tars. Other accounts in Hans' documents detail consumption of beer and bread, with higher-quality varieties differentiated according to rank and social status [38]. Higher-ranking individuals ate bread and biscuits made from wheat and washed it down with German beer. The working crew endured hard bread and lower quality local beer. Future excavations on *Gribshunden* may reveal even more stark differences in the food and drink of the nobility and the common soldier and sailor.

Baltic underwater archaeobotany offers additional victualing details from other medieval shipwrecks. Finds on the 15[th] century Copper Wreck in the gulf of Gdansk included food remains of beans, plums, hazel, walnuts and onions [16]. The wreck also contained garlic, interpreted as medicinal [17]. These archaeobotanical data complement information in documents of the ship's owner. These list expenses for victuals including meat, fish, salt, butter, fat,

bread, flour, and beer [16]. Very few other medieval ships from the Baltic Sea have been archaeobotanically investigated. Two 13ᵗʰ century wrecks are the Gedeby shipwreck, by the island of Falster, Denmark; and a cog vessel close to Oskarshamn, Sweden [127,128]. These were small vessels transporting livestock, grain, and iron. The wrecks produced only a few cereal food plants, interpreted as cargo remains.

Outside the Baltic region, a well-documented wreck from Northern Europe is the 16ᵗʰ century carrack-type warship *Mary Rose*, flagship of Henry VIII [15]. Botanical remains from the wreck include peppercorn, hops, cereals, hazel, walnut, plum/greengage, cherry, and grapes. While nuts and fruit would have added variety to the diet, it is unclear if these were limited to officers, particularly as the fruits were mainly found in the ship's hold and some in the galley supplies. The presence of peppercorns on *Mary Rose* is of particular interest for the study of *Gribshunden*. The recovery of peppercorn from multiple loci on *Mary Rose* suggests that some could have been for general use in the galley, while individual crew members also cached some for their personal use [129]. Among the contents of a barbers-surgeon's chest were a few peppercorns, believed to have been part of medicinal supplies. Future excavations on *Gribshunden* may reveal similar multiplicity of use.

The plant material from *Gribshunden* contributes new knowledge about the foodstuffs consumed by the social elite in medieval Scandinavia. Considering that *Gribshunden* sank in the beginning of June, perishables such as ginger, grapes, berries, and cucumber were likely preserved as dried fruit, pickles, or jams to have been available for consumption all year around. It is unclear if ginger rhizomes were stored fresh or were preserved in some form. If fresh, the rhizomes must have been procured within days of *Gribshunden*'s departure from Copenhagen, as fresh ginger has a short shelf life. Other foodstuffs recovered from *Gribshunden* could be stored for far longer than fresh ginger. Spices from far distant origin, such as black pepper, saffron, and cloves would keep for long periods if they remained dry. Dill, black mustard, and caraway were likely sourced locally. Flaxseeds, almond, and hazelnut have long storage lives. It is probable that nuts were stored on board in their shells and cracked opened when ready for consumption, as broken shell parts were recovered from both nut species.

It is tempting to compare this wide variety of fresh produce to records of medieval maritime provisioning; but as the royal flagship, *Gribshunden* is a special case. Instead, the exotic foodstuffs from the king's spice cabinet provide a window into the consumption patterns that likely followed in the elite landscapes of castles ashore. Despite the popularity of exotic spices among the medieval aristocracy, very few of these foods have survived archaeologically. The preservation of these plant foods on *Gribshunden* constitutes a discovery of great historical value. Spices and other exotic foods such as almonds were typically consumed only by society's wealthiest. On *Gribshunden* these were not victuals for the working crew. Exotic food items are probably some of the most easily identifiable indicators of social context. King Hans was travelling on the ship together with his courtiers; these expensive exotic foods are linked to these passengers.

Danish archival sources from 1487 relate brief but telling details specific to King Hans' expenses and activities aboard his flagship [38]. While laid up awaiting favorable winds in 1487 en route to Gotland and at a stop on Bornholm island on the return, Hans gambled on card games. In those few weeks, his recorded losses totaled 42 marks, nearly the annual salary of one of the ship's senior officers. He ate candy and nuts, and with his companions, drank wine and particularly beer. On that voyage the ship reprovisioned with fresh barrels of local beer, as well as *embstøll*, a hopped Prussian beer originally brewed in Einbeck, Germany. Other recorded purchases for Hans' sea voyages are consistent. He bought more confectionaries for the apothecary, nuts, and saffron while voyaging to Års, Jylland, Denmark. The amount of saffron purchased was prodigious: the cost was 36 mark danske, equivalent to nine months

of salary for a senior officer on *Gribshunden*, or 18 months of salary for a sailor [38]. These documentary references combined with the remains of saffron, almonds, and hazelnut recovered from *Gribshunden*'s 1495 wrecking prove that the king regularly consumed these extravagant foods while at sea, and most probably while ashore.

In addition to information about the specific foodstuffs consumed, the written sources reveal circumstantial information about procedures and etiquette related to the king's elite milieu at sea. The *Gribshunden* accounts of 1493 list royal court employees [38]. The "*kelderswen*" (källarsven), perhaps similar to an army quartermaster, managed the food stores. The "*dugeswen*", perhaps similar to a chief steward, was responsible for laying out the royal tables for meals and banquets. No doubt similar roles and procedures were spelled out and followed when King Hans traveled on his flagship two years later.

## The social context of luxury foods and spices

Food, both in its acquisition and consumption, provides insights to the economic and social aspects of a society. The archaeobotanical assemblage from *Gribshunden* shines a light on late medieval Nordic social structure. As King Hans traveled aboard his flagship on its final voyage, accompanied by a fleet of perhaps 18 other vessels, different social groups walked those decks. The king represented the pinnacle of the social hierarchy, the royal court and nobles represented the elite, the ship's captain and senior officers another stratum, and the hired soldiers and ship's crew the lowest social group in this floating castle. All of these individuals played important roles in the pageantry of the Kalmar political summit.

Saffron, black pepper, clove, ginger, almonds and grapes were exotic foods and spices out of reach for most people in medieval society due to cost and rarity. Only those who possessed sufficient financial resources could afford these expensive food items, a luxury limited to the elites. The elite certainly enjoyed consuming these foods, but their high cost and exotic nature also marked their social status [130].

The role of luxury foods and spices on *Gribshunden* may have served different purposes. The types of luxury foods found were primarily those of flavorings, and while at sea, these could have had been used for preparation of meals served to the King and his men, and other dried foods (grapes and nuts) could have been enjoyed as snacks. An interesting analog is a later royal document from 1627, which show that Gustav II Adolf (King of Sweden 1611–32) was well-stocked with spices and other exclusive foods on his naval trips [131]. Cucumbers, almonds, rice, figs, ginger, pepper, anise, nutmeg, saffron and alum were on his menu. Such historic documentation together with the botanical food remains from *Gribshunden*, suggest that consumption of luxury foods among royals was not limited to the main living quarters on land, but was part of the royal environment at all times, even at sea. Food is more than diet. It is inextricably linked to the expression of social and cultural identity. The consumption of luxurious foods was a crucial aspect of King Hans' and other monarchs' presentation of themselves to their subjects.

Had *Gribshunden* safely arrived in Kalmar, from its decks Hans would have employed all manner of elite signaling to impress the Swedish Council. The consumption of exotic foods certainly was symbolic of prestige and social superiority within Hans' realm. It also demonstrated that King Hans and medieval Denmark were culturally integrated with the rest of Europe, and the world beyond the continental borders. The goal of the Kalmar negotiations was for Sweden to re-join the Nordic Union, consolidating the entire region under Hans' rule. During the summer-long summit, Hans would have hosted banquets for the Swedish Council, possibly onboard *Gribshunden* but more likely on the grounds of Kalmar Castle and within the castle itself. Through hospitality and feasting, food is recognized in many cultures as a tool for

the creation and maintenance of social relations. Feasting embodies two principal characteristics: first, a communal aspect of consumption in large groups, with food (and drinks) that are different from everyday practices; and second, a display of social prominence, with the presentation of food conveying economic success, high status, and cultural power [130]. Additionally, knowledge from and of foreign cultures is a way for the élites in stratified societies to manifest their status and secure political power [132]. All of the events that might have procceded in Kalmar were likely constructed not merely for practical purposes; they were political theater. The presentation of exclusive foods and decorative dishes would, in both organization and content, be designed as an integral part of the entertainment, and be a part of the display of the king's status and power.

The banquets hosted by the elites featured feasting and lavish displays. These established and strengthened relationships, and were critical activities to bind political support and networks between states [133,134]. An example of this from the late medieval period is the great feast hosted by Henry VIII at Greenwich in 1527 to celebrate the conclusion of a treaty with France. From written accounts of the event, the courtly feast displayed a variety of luxurious foods and featured several exotic plant foods, such as clove, ginger, almond, mace, and cinnamon (but no saffron) [108]. If King Hans planned a banquet, the use of exotic foods and spices in elaborate cuisine would have showcased the symbolic power of the Danish crown. It would have been clear to all witnesses and participants that the Danish court followed the culinary trends of European courtly societies. It would have signaled the cosmopolitan identity of the king, his access to luxury commodities from far distant origins, and his royal court's position in the global economy.

Along with hosting feasts, gift-giving played an integral part in the building and strengthening of relations in medieval culture [135]. Gifts of food and drink were common across medieval society. They could be contributions in kind to feasts or celebrations, or take the form of support and donations to individuals or institutions [136,137]. These gifts were as significant as the rank of the donor. In the upper echelon of northern European society, the range of gifted foodstuffs could be venison or other game, high-status birds, fresh fish, and wine [136]. Gifts played an important role in negotiations as well. The role of spices as prestige commodities during the Middle Ages was underlined by their use as gifts on political occasions [135]. For instance, spices prepared as gifts for ambassadors in the Baltic region are documented from Reval [46]. In 1494, the Reval Town Council sent an ambassador to Moscow. As gifts to dukes and a bishop, he presented foodstuffs including boxes of spices. On another occasion when traveling to meet nobles in Wenden and Narva in 1498, among the foods presented by the ambassador from Reval were pepper, saffron, ginger, and mustard. *Gribshunden*'s luxury foods similarly might have been intended as representation gifts on arrival in Kalmar. Similar to feasts, gifts made by the crown would express an opportunity to secure the peace by strengthening a personal relationship between the two rulers. Gifts, banquet ingredients, or at-sea consumables: given the context and the circumstances of the journey, all are possible purposes for the exotic foods recovered from *Gribshunden*.

## Conclusion

This first study of plant remains from the 15[th] century flagship *Gribshunden* demonstrates that delicate plant parts that normally do not survive in the archaeological contexts on land can persist in the marine environment. The archaeological potential of plant remains in underwater environments connected to wreck sites is extraordinary. The presence of a range of economic plants aboard the ship indicates a high-status lifestyle involving the use of luxury foods. The exceptional finds of saffron, clove, ginger, black peppercorn, and almond from the

shipwreck offer a unique keyhole view to the historical consumption and social signifying of luxurious foods. Access to these items was restricted by high cost. On *Gribshunden* as on shore, the possession and consumption of such luxury plants would be inextricably linked to the most elite social milieu.

## Supporting information

**S1 Table. List of archaeobotanical plant remains from the *Gribshunden* wreck site.** (XLSX)

## Acknowledgments

The authors thank Blekinge Museum Director Marcus Sandekjer and Head of Collections Christoffer Sandahl. We acknowledge the 2021 archaeological field team: Jan-Erik Andersson, Staffan von Arbin, Mikael Björk, Paola Derudas, Paolo Iglic, Marie Jonsson, and Phillip Short. We thank Marcus Lecaros for image editing. The authors thank the academic editors and anonymous reviewers for their constructive comments and suggestions.

## Author Contributions

**Conceptualization:** Mikael Larsson, Brendan Foley.

**Data curation:** Mikael Larsson, Brendan Foley.

**Formal analysis:** Mikael Larsson.

**Funding acquisition:** Mikael Larsson, Brendan Foley.

**Investigation:** Mikael Larsson, Brendan Foley.

**Methodology:** Mikael Larsson, Brendan Foley.

**Project administration:** Mikael Larsson, Brendan Foley.

**Resources:** Mikael Larsson, Brendan Foley.

**Supervision:** Mikael Larsson.

**Validation:** Mikael Larsson.

**Visualization:** Mikael Larsson, Brendan Foley.

**Writing – original draft:** Mikael Larsson, Brendan Foley.

**Writing – review & editing:** Mikael Larsson.

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
