## [Decision Letter · Decision Letter 0]

25 Nov 2022

PONE-D-22-28990The king’s spice cabinet – Plant remains from Gribshunden, a 15th century royal shipwreck in the Baltic SeaPLOS ONE

Dear Dr. Larsson,

Thank you for submitting your manuscript to PLOS ONE. After careful consideration, we feel that it has merit but does not fully meet PLOS ONE’s publication criteria as it currently stands. Therefore, we invite you to submit a revised version of the manuscript that addresses the points raised during the review process. Both reviewers make suggestions for improving the manuscript. Please consider these while making your revisions, and pay particular attention for the need of additional references for factual statements. Text citation numbers need to be in brackets rather than parentheses. Please be sure to check the references cited section to make sure it is consistent with the journal's style requirements. Also please provide accession or catalog numbers for the specimens if available.

We look forward to receiving your revised manuscript.

Kind regards,

John P. Hart, Ph.D.

Academic Editor

PLOS ONE

Journal Requirements:

2. In your manuscript, please provide additional information regarding the specimens used in your study. Ensure that you have reported specimen numbers and complete repository information, including museum name and geographic location. If permits were required, please ensure that you have provided details for all permits that were obtained, including the full name of the issuing authority, and add the following statement:

For more information on PLOS ONE's requirements for paleontology and archaeology research, see https://journals.plos.org/plosone/s/submission-guidelines#loc-paleontology-and-archaeology-research.

4. We note that Figures 1 and 2 in your submission contain [map/satellite] images which may be copyrighted. All PLOS content is published under the Creative Commons Attribution License (CC BY 4.0), which means that the manuscript, images, and Supporting Information files will be freely available online, and any third party is permitted to access, download, copy, distribute, and use these materials in any way, even commercially, with proper attribution. For these reasons, we cannot publish previously copyrighted maps or satellite images created using proprietary data, such as Google software (Google Maps, Street View, and Earth). For more information, see our copyright guidelines: http://journals.plos.org/plosone/s/licenses-and-copyright.

a. You may seek permission from the original copyright holder of Figures 1 and 2 to publish the content specifically under the CC BY 4.0 license.  

Reviewers' comments:

Reviewer's Responses to Questions

**Comments to the Author**

1. Is the manuscript technically sound, and do the data support the conclusions?

Reviewer #1: Yes

Reviewer #2: Yes

2. Has the statistical analysis been performed appropriately and rigorously? 

Reviewer #1: Yes

Reviewer #2: N/A

3. Have the authors made all data underlying the findings in their manuscript fully available?

Reviewer #1: Yes

Reviewer #2: Yes

4. Is the manuscript presented in an intelligible fashion and written in standard English?

Reviewer #1: Yes

Reviewer #2: Yes

5. Review Comments to the Author

Reviewer #1: This paper present important evidence that should be published, however, I think on occasion the text can be repetitive and so I suggest the authors go through and check whether text is being repeated and either condense or remove. For example, I think that the section 'The Baltic preservative environment and Northern European maritime archaeobotanical comparanda' (98) although useful is repeated in more detail and with more references in the discussion, making this paper very long while not adding any new information. I would suggest removing this and just commenting in the introduction that these sites exist and that you will be looking at them more in detail in the discussion. The section on preservation (100-112) could then go into the intro to explain the site.

I also feel that the authors should double check where they reference and where they don’t. Periodically I wanted to see where they got that information from and there didn’t seem to be a reference e.g. around 778 (the archival sources).

Table two I think needs some more information. First, how many sites are we talking about? You could have frequency rather than presence absence, because we don’t know how rare the finds are from this table. The figures look great, although Figure 2 could be clearer as it was a bit pixelated in my version.

Reviewer #2: The manuscript presented for the review addresses an interesting aspect of maritime archaeobotany. It should be stressed that analysis of plant remains from underwater sites are among the less frequently described, especially in the context of Baltic region, which is related to the nature of the site. Some of the plant findings may be overlooked at the stage of collecting samples, and some of them are damaged due to inadequate storage or conservation processes. In this context, very careful work is essential, as it means - undamaged samples of the material and its thorough preparation for research. This made it possible to add more information, this time botanical, about the history of Gribshunden, a 15th century royal shipwreck. The very interesting element of the article is the comparison of the obtained results with archaeobotanical and historical data from not only ship wrecks, but also from Baltic towns.

A huge contribution of the article is the presentation of archaeobotanical data, which are generally not recorded at terrestrial archaeological sites. First of all I mean the finds of saffron, the presence of which in ancient towns is generally known only on the basis of historical sources

Generally, the submitted article is very well suitable for publication in PlosOne as the data set and the available information is impressive, the research methods are a good example of interdisciplinary cooperation and the historical background for data interpretation is comprehensively enrolled by the article. I recommend the manuscript to be accepted after minor revision.

In detail I want to suggest the following changes or I have some questions. I did not add these comments to the text because it seems to me that what she presented below is understandable.

1. The overall structure of the article is correct, but in my opinion, the Abstract should be shortened. I think that too much historical data was presented here, which is important, but not quite in this part of the article. I would put more emphasis on pointing out the uniqueness of the archaeobotanical finds described in the article.

2. Introduction is also rather lengthy. It contains a general background on the history of the ship and archaeobotanical study. The research problem is only presented at the end of this long part. I suggest a small reorganization and focusing on the main research problem.

3. Materials and methods - According to the Authors, the botanical material was preserved uncharred. Could any part of the plant remains be partially mineralized?

4. The chapter The Baltic preservative environment and Northern European maritime archaeobotanical comparanda it seems like a good part for discussion. I suggest to move it.

5. Table 1. List of economic plant species… – I believe that taxa such as Rubus spp. should be in the fruit category. Nuts should be a separate category.

6. Fig. 1. I would prefer that the map shows the main cities within the Baltic Sea where archaeobotanical research was conducted and the results of which were used to create the article.

7. I wonder if it's not worth commenting on the lack of figs (Ficus carica) in the materials from the wreck. Fig remains are one of the most commonly found traces of exotic fruit in materials from mediaeval towns. According to the authors, is her absence from the list due to the possibility of being overlooked or is she simply not on the ship? Which would probably be strange considering the presence of other luxury products.

8. Flax (Linum usitatissimum) it could also mean a kind of medicinal plant.

9. Henban (Hyoscyamus niger) represents ruderal weeds. Of course, it is also a very interesting plant which is used in traditional herbal medicine. I would be very careful with a story about the plant's role on a royal ship based on a single find, which could be totally accidental. Yes, Authors wrote “If future excavation on the wreck produces greater numbers of henbane seeds, its purpose may become clear”, but I suggest shortening this paragraph.

10. Copy editing of the English is needed as some phrases seem to be too long and too complex (hard to read at times…).

6. PLOS authors have the option to publish the peer review history of their article (what does this mean?). If published, this will include your full peer review and any attached files.

Reviewer #1: No

Reviewer #2: No

---

## [Author Response · Author response to Decision Letter 0]

3 Jan 2023

We are grateful for the comments and suggestions by the reviewers and appreciate the time taken to do so. The suggestions have been very useful and have much improved the submitted manuscript. We have followed most of the recommendations and suggestions by the reviewers. All changes can be tracked in the Revised Manuscript with Track Changes. 

We look forward to hearing from you regarding our submission. We will respond to any further questions and comments that you may have.

Thanks for your time!

Author comments on revision

Specific corrections to the manuscript are separated below according to sections. More generally, as suggested by reviewer #2, the language and phrasing has been revised throughout the manuscript. In addition, repetitive text has been either condensed or removed. 

Abstract

Reviewer #2 pointed out that the abstract was rather long and had too many historical details. We wish, however, for the abstract to remain as first submitted. If this study was intended for an archaeobotanical journal we would condense the historical aspect. As this is an interdisciplinary study, with potential interest among readers from various research fields, we find that providing details on the historical context of the botanical finds to potentially be of interest when reading the abstract. 

Word limit according to the journal is 300 words, our abstract is 268 word. 

Introduction

Both reviewers commented on the Introduction section being rather lengthy. To address this, the Introduction section was reorganized. The two subheadings Historical background and The Baltic preservative environment and Northern Europe maritime archaeobotanical comparanda were merged and some text was removed. The research focus is more explicitly emphasized as well. Some historical data was removed as these aspects of the study is discussed in more detail in the Discussion section. As advised by reviewer #1, the part on preservation was placed in the Introduction to explain the site. Subheading Archaeological background was removed from the Introduction, and placed as a separate heading following the Introduction.

Material and methods

All material recovered were subfossil. Partial mineralization of plant remains was not observed. 

Results

The absence of fig seeds in the samples is indeed, noteworthy. As pointed out by reviewer #2, seeds from figs is a common exotic food commodity in material from medieval towns. All samples were carefully analyzed and no botanical remains or species were overlooked. Fig seeds were simply not in the collected samples. However, future investigations of the shipwreck may present new species and complement the economic food plants presented in this study. 

Discussion

The five food categories was revised as suggested by reviewer #2. Raspberry and blackberry has been placed under the category Fruits and vegetables, and the two nut species (almond, hazel) are placed in a separated category, Nuts. This reorganization is corrected throughout the manuscript.

Subheading Oilseeds. That flax may have had medicinal application has been commented. 

Subheading Medicinal plants. As pointed out by reviewer #2, while henbane is an interesting plant, having served multiple purposes in medieval times, with only one seed found, it is somewhat unclear if it representing medicinal use or a ruderal weed. We therefore, as suggested by reviewer #2, shortened the discussion on henbane. 

Figures

Figure 1. As suggested by reviewer #2, the map was edited to include locations of the main medieval towns along the Baltic coasts with archaeobotanical data discussed in the manuscript.

Figure 2. Resolution of figure 2 may have been affected by uploading the submission as a pdf when sent to reviewers.

Tables

Table 1. Plant categories was adjusted as suggested by reviewer #2. Berries are placed in Fruits and berries; nuts is a category on its own. 

Table 2. To add sites and frequency of recorded number of seeds rather than presence and absence is a good point raised by reviewer #1. Based on the finds from Gribshunden, the purpose of the table was, however, for the reader to get a visual overview of the presence of these species found in the archaeological layers of medieval towns in Scandinavia and around the Baltic region. To include sites and seed frequency in the table is problematic for some species and geographical areas. As some original sources presenting archaeological data from the Baltic region is only reported or published in the native language of respective countries of the region, or reported in only domestic journals or books, we have had to use secondary literature for some archaeobotanical information. Thus, to list frequency for some species and only presence for others would be inconsistent. For this reason, we decided to keep presence and absence in Table 2 as originally submitted. Nevertheless, in the subsection Economic plants from the Gribshunden shipwreck and archaeological comparanda of the Discussion section, we discuss sites or town from where botanical remains of respective species have been found, and if botanical remains for the species is common or rare, or when possible, we mention frequency. 

As dill, black mustard, caraway are common in cultural layers around Scandinavia and the Baltic region during the Middle Ages, these species were removed from table 2. Instead, we focus on the plants species found at the wreck site of Gribshunden that are rare and exotic in Scandinavia and the Baltic region. 

References

References in the text have been double check to make sure that references are placed correctly. 

Seven references were added to the manuscript:

Kamatou, GP, Vermaak, I, Viljoen, M, 2012. Eugenol - from the remote Maluku Island to the international market place: a review of a remarkable and versatile molecule. Molecules 2012, 17, 6953-6981

Etting V, Straetkvern K, Gregory D. Gribshunden. In: Nationalmuseets Arbejdsmark 2019. 2019. p. 102–13.

Franckenius, Johansson 1638(1659): Speculum botanicum. 2nd edition, Upsaliae

Thangaselvabal T., Gailce Leo Justin C., Leelamathi M. Black pepper (Piper nigrum L.) ‘the king of spices’ – A review. Agricultural reviews, 2008, vol 29, issue 2, 89-98.

Hooker J.D. 1886. The Flora of British India. L. Reeve and Co., London, pp. 78–95. 5.

Corbineau, R, Ruas, M-P, Barbier-Pain, D, Fornaciari, G, Dupont, H, Colleter, R. Plants and aromatics for embalming in Late Middle Ages and modern period: a synthesis of written sources and archaeobotanical data (France, Italy). Vegetation History and Archaeobotany, 2018, 27 (1), pp.151-164. https://doi.org/10.1007/s00334-017-0620-4

Golaris A. Saffron cultivation in Greece. In: Negbi M, editor. Saffron : Crocus sativus L. Amsterdam, Harwood Academic Publishers, 1999, p. 73-85

Two references were removed from the manuscript:

Lempiäinen M, Timonen T, Harju P, Alvik R. Underwater archaeobotany: plant and wood analyses from the Vrouw Maria, a 1771 shipwreck in the Finnish Baltic Sea. Veg Hist Archaeobot [Internet]. 2022;31(1):97–106. Available from: https://doi.org/10.1007/s00334-021-00840-3

Manders M, Kuijper W. Shipwrecks in Dutch Waters with Botanical Cargo or Victuals. In: Bakels C, Kamermans H, editors. Analecta Praehistorica Leidensia. Leiden: Leiden University; 2015. p. 141–72.

Additional requirements outlined by the journal.

1. Style requirements have been addresses for the title page.

2. Figure 1 and 2; format has been change from Pdf to tiff. 

3. Table 1 and 2; reformatted to make them editable. In doing so, Table 2 had to be changed from landscape view to portrait view to meet journal requirements.

4. Reference list has been reviewed to make sure that it is complete and correct. 

5. Repository information and permit numbers (incl. full name of the issuing authority, have been added to the section: Material and Method.

6. Funding information has been revised.

7. Copyright details has been added to the figure caption to Figs 1 and 2 in the manuscript and a Content Permission Form has been uploaded from illustrator Frida Nilsson at Lund University.

---

## [Editor Report · Decision Letter 1]

13 Jan 2023

The king’s spice cabinet – Plant remains from Gribshunden, a 15th century royal shipwreck in the Baltic Sea

PONE-D-22-28990R1

Dear Dr. Larsson,

We’re pleased to inform you that your manuscript has been judged scientifically suitable for publication and will be formally accepted for publication once it meets all outstanding technical requirements.

Kind regards,

John P. Hart, Ph.D.

Academic Editor

PLOS ONE
---

## [Editor Report · Acceptance letter]

17 Jan 2023

PONE-D-22-28990R1 

The king’s spice cabinet – Plant remains from *Gribshunden*, a 15^th^ century royal shipwreck in the Baltic Sea 

Dear Dr. Larsson:

I'm pleased to inform you that your manuscript has been deemed suitable for publication in PLOS ONE. Congratulations! Your manuscript is now with our production department. 

Kind regards, 

on behalf of

Dr. John P. Hart 

Academic Editor

PLOS ONE